# Regulation of RNA granule dynamics by phosphorylation of serine-rich, intrinsically disordered proteins in *C. elegans*

**Jennifer T Wang[1], Jarrett Smith[1], Bi-Chang Chen[2,3], Helen Schmidt[1], Dominique Rasoloson[1], Alexandre Paix[1], Bramwell G Lambrus[1], Deepika Calidas[1], Eric Betzig[3], Geraldine Seydoux[1]***

[1]Department of Molecular Biology and Genetics, Howard Hughes Medical Institute, Johns Hopkins University School of Medicine, Baltimore, United States; [2]Research Center for Applied Sciences, Academica Sinica, Taipei, Taiwan; [3]Janelia Farm Research Campus, Howard Hughes Medical Institute, Ashburn, United States

**Abstract** RNA granules have been likened to liquid droplets whose dynamics depend on the controlled dissolution and condensation of internal components. The molecules and reactions that drive these dynamics in vivo are not well understood. In this study, we present evidence that a group of intrinsically disordered, serine-rich proteins regulate the dynamics of P granules in *C. elegans* embryos. The MEG (maternal-effect germline defective) proteins are germ plasm components that are required redundantly for fertility. We demonstrate that MEG-1 and MEG-3 are substrates of the kinase MBK-2/DYRK and the phosphatase $PP2A^{PPTR-\frac{1}{2}}$. Phosphorylation of the MEGs promotes granule disassembly and dephosphorylation promotes granule assembly. Using lattice light sheet microscopy on live embryos, we show that GFP-tagged MEG-3 localizes to a dynamic domain that surrounds and penetrates each granule. We conclude that, despite their liquid-like behavior, P granules are non-homogeneous structures whose assembly in embryos is regulated by phosphorylation.

*For correspondence:
gseydoux@jhmi.edu

## Introduction

RNA granules are ubiquitous cytoplasmic organelles that contain RNA and RNA-binding proteins (*Kedersha et al., 2013*; *Buchan, 2014*). Several types of RNA granules have been described, including germ granules in germ cells, neuronal granules in neurons, and stress granules and P bodies in somatic cells. Their functions include mRNA transport and storage and the regulation of mRNA degradation and translation. Unlike other organelles, RNA granules are not bound by limiting membranes and their internal components are in constant flux with the surrounding cytoplasm. RNA granules assemble and disassemble in response to developmental or environmental cues (*Kedersha et al., 2013*; *Buchan, 2014*). Live imaging studies in *Caenorhabditis elegans* zygotes have suggested that RNA granules behave like liquid droplets that undergo phase transitions (*Brangwynne et al., 2009*). Granule components exist in a condensed liquid or gel-like phase in the granule and a dispersed phase in the cytoplasm (*Weber and Brangwynne, 2012*; *Toretsky and Wright, 2014*). In vitro studies have lent support to this hypothesis by demonstrating that purified proteins can undergo phase transitions in aqueous solutions. For example, proteins that contain weak, multivalent protein-binding domains undergo liquid–liquid demixing in concentrated solutions to form micron-sized droplets (*Li et al., 2012*). Similarly, RNA-binding proteins that contain low sequence-complexity domains assemble into ordered fibers that coalesce into hydrogels when maintained at low temperature (*Frey et al., 2006*; *Kato et al., 2012*). The proteins that drive the phase transitions in vivo, however, are not known.

**eLife digest** For a gene to be expressed as a protein, its DNA is first used as a template to make a molecule of RNA, which is then translated to make the protein. In most cells, RNA molecules concentrate into aggregates called RNA granules. These granules contain both RNA and proteins that bind to RNA and are used to transport, store, and regulate the translation and breakdown of RNA molecules. Unlike many other structures within cells, RNA granules are not surrounded by a membrane; and the molecules that hold RNA granules together are not known.

P granules are a type of RNA granule that is found in the germ cells (the cells that go on to form eggs and sperm) of a microscopic worm called *C. elegans*. When a *C. elegans* embryo is still a single cell, P granules move throughout the cell and the P granules at the front of the cell dissolve, while those at the back condense. As such, when the single-celled embryo divides, the front forms a cell without P granules (that will go on to form the tissues of the worm's body) and the back becomes a P granule-containing germ cell.

Two proteins called MBK-2 and PPTR-1 have opposite effects on P granules: MBK-2 causes P granules to dissolve, while PPTR-1 makes them form. MBK-2 is an enzyme that adds phosphate groups onto other proteins, whereas PPTR-1 is part of an enzyme that removes such groups. Wang et al. have now searched for proteins that interact with MBK-2 and PPTR-1 in order to identify the molecules that regulate the assembly of P granules. They found that a group of proteins, known as MEG proteins, are acted upon by both of these proteins. Wang et al. found that MBK-2 adds phosphate groups to MEG proteins, which encourages granules to disassemble, while PPTR-1 removes these groups to promote granule assembly.

Wang et al. generated mutant worms that lacked each of the MEG proteins. These mutant worms had fewer and smaller P granules than normal worms. Without MEG proteins, P granules failed to assemble or disassemble normally and the worms were infertile. Using high resolution microscopy, Wang et al. observed that the MEG proteins wrap around the P granules and that one of the MEG proteins—called MEG-3—follows an almost ribbon-like path that surrounds and enters each granule. These observations suggest that the MEG proteins stabilize RNA granules by forming a cage-like scaffold around each granule. How the MEG proteins—which are predicted to lack a fixed or ordered three-dimensional structure and show no similarity to proteins with known functions—assemble into a scaffold will be the focus of future studies.

Several proteins and RNAs are required for granule integrity (*Buchan, 2014*), and recently the kinase DYRK3 has been implicated in the regulation of stress granule dissolution and condensation (*Wippich et al., 2013*). In the present study, we identify the substrates of the *C. elegans* DYRK3 homolog MBK-2 and demonstrate that phosphorylation and dephosphorylation of these substrates drive the dissolution and condensation of P granules in embryos.

P granules are the germ granules of *C. elegans* (*Strome and Wood, 1982*; *Updike and Strome, 2010*). P granules are perinuclear during most of the germline development. During the oocyte-to-embryo transition, P granules detach from nuclei and disperse in the cytoplasm for asymmetric segregation to the nascent embryonic germline. Preferential dissolution of the granules in the anterior and condensation in the posterior of the zygote cause the granules to accumulate in the cytoplasm destined for the germline blastomere $P_1$ (*Figure 1*) (*Brangwynne et al., 2009*; *Gallo et al., 2010*). P granules segregate asymmetrically for three more divisions until the P granules are uniquely concentrated in the germline founder cell $P_4$ (*Strome and Wood, 1983*). Two groups of RNA-binding proteins form the core components of P granules: the RGG domain proteins PGL-1 and PGL-3 and the Vasa-related helicases GLH-1, 2, 3, and 4 (*Updike and Strome, 2010*). A self-association domain in the PGL family is essential for granule nucleation (*Hanazawa et al., 2011*), and FG repeats in GLH-1 are required for P granules to associate with nuclei (*Updike et al., 2011*).

In previous studies, we identified two potential regulators of P granule dynamics: MBK-2, the *C. elegans* DYRK3 homolog (*Aranda et al., 2011*; *Wippich et al., 2013*) and PPTR-1, a regulatory subunit of the heterotrimeric phosphatase PP2A (*Padmanabhan et al., 2009*). Loss of *mbk-2* and *pptr-1* has opposite effects on P granule dynamics. All P granules remain condensed in *mbk-2* zygotes and all P granules disperse in *pptr-1* zygotes (*Pellettieri et al., 2003*; *Quintin et al., 2003*; *Gallo et al., 2010*).

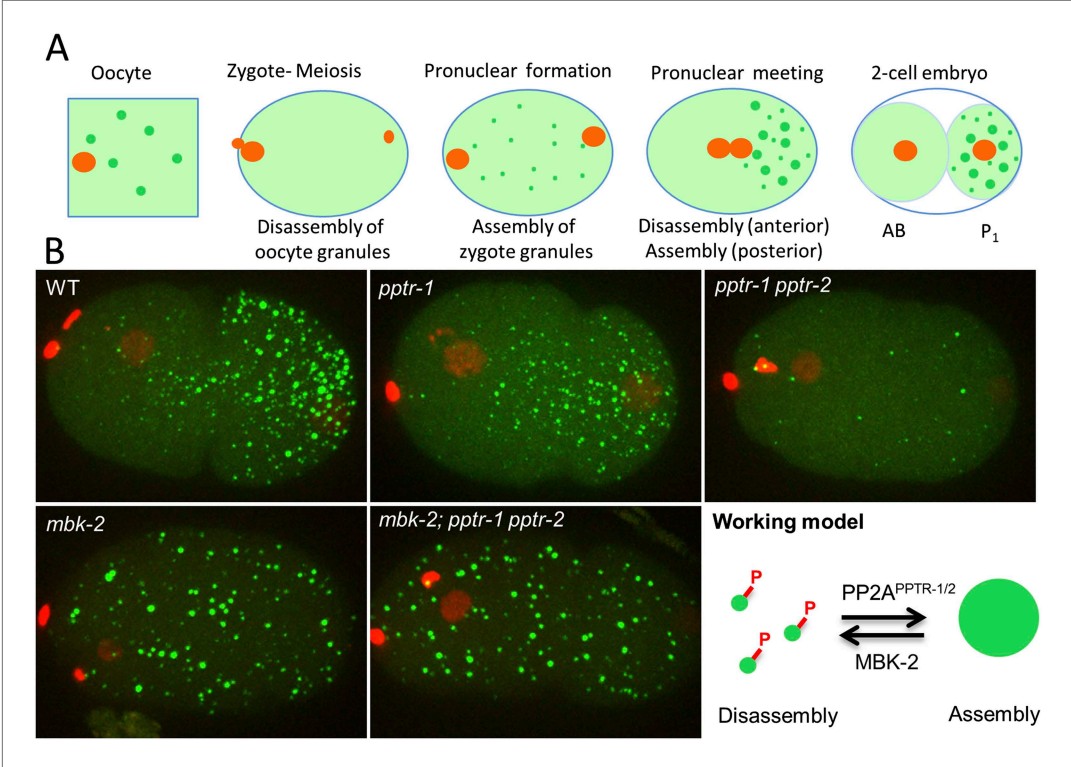

**Figure 1**. MBK-2 and PP2A$^{PPTR-1/2}$ are an opposing kinase/phosphatase pair. (**A**) P granule dynamics during the oocyte to embryo transition. Green puncta are P granules, pale green color represents P granule components that diffuse into the cytoplasm. Orange represents pronuclei. P$_1$ is the germline blastomere. (**B**) Zygotes of the indicated genotypes expressing GFP::PGL-1 and mCherry::Histone 2B during pronuclear migration. Full genotypes are *pptr-1(tm3103)*, *pptr-1(tm3103) pptr-2(RNAi)*, *mbk-2(RNAi)*, and *mbk-2(RNAi);pptr-1(tm3103) pptr-2(RNAi)*. Right: working model. Phosphorylation by MBK-2 disassembles granules. Dephosphorylation by PP2A$^{PPTR-1/2}$ assembles granules. Each embryo is 50 µm in length. Anterior is to the left, posterior is to the right.

The following figure supplement is available for figure 1:

**Figure supplement 1**. Oocyte P granule disassembly during the oocyte-to-zygote transition.

These phenotypes suggest that MBK-2 and PPTR-1 proteins may share substrates whose phosphorylation regulates the phase of P granules. To investigate this possibility, we searched for MBK-2 and PPTR-1 substrates and identified a group of intrinsically disordered proteins that regulate P granule dynamics.

## Results

### PPTR-1 functions redundantly with PPTR-2 to stabilize P granules and is antagonized by MBK-2

To determine the earliest time point at which P granules become dynamic during the oocyte-to-zygote transition, we monitored P granules in the oviduct and uteri of live hermaphrodites from ovulation through the first embryonic divisions. We used a GFP::PGL-1 transgene to mark P granules and an mCherry::histone transgene to mark chromosomes. Shortly after ovulation, P granules that were present in the oocyte cytoplasm (oocyte granules) disassemble and new granules (zygote granules) form throughout the cytoplasm as the maternal pronucleus completes the second meiotic division (*Video 1*, *Figure 1A*, *Figure 1—figure supplement 1*). After meiosis, as the zygote becomes polarized along the anterior/posterior axis, granules in the anterior cytoplasm are quickly disassembled and granules in the posterior cytoplasm continue to grow and fuse to form large (≥1 micron) granules (*Figure 1*, *Video 2*). We observed the same dynamics with a mCherry fusion to a second P granule component (PGL-3) (*Figure 1—figure supplement 1*).

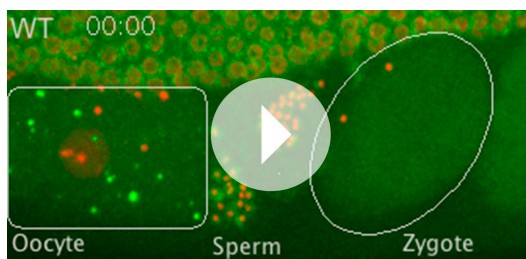

**Video 1**. P granule dynamics during the oocyte-to-zygote transition. Time-lapse of eggs undergoing the oocyte-to-embryo transition in the gonad of a hermaphrodite expressing GFP::PGL-1 and mCherry::H2B. An oocyte in the oviduct, several sperm in the spermatheca, and a zygote in the uterus are highlighted in the first frame of the movie. Key stages in the P granule disassembly and assembly process are also highlighted in later frames. Images are maximum intensity projections of 11 z stacks separated by 1 µm steps. Stacks were taken every 30 s, total movie time is 30 min 30 s, movie is played back in 60× real time.

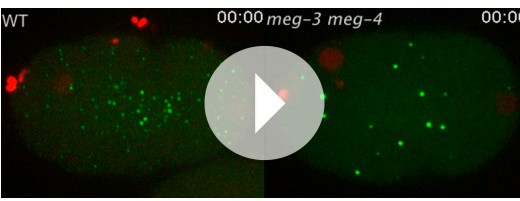

**Video 2**. P granule dynamics in wild-type and meg-3 meg-4 zygotes. Time-lapse of embryos expressing GFP::PGL-1 and mCherry::H2B. Images are maximum intensity projections of 8 z planes separated by 1 µm steps. Stacks were taken every 8 s, total movie time is 17 min 52 s, movie is played back in 80× real time. Full genotype of mutant is *meg-3(tm4259) meg-4(ax2026)*. Very small granules form transiently in the posterior of the *meg-3 meg-4* embryo at 7:20. The bright red puncta in the anterior (left) side of the embryos are polar bodies. In the wild-type movie, there are also red puncta above the embryo, these are sperm outside of the egg shell.

The kinase MBK-2 is required for P granule asymmetry in zygotes (*Pellettieri et al., 2003*; *Quintin et al., 2003*). We found that in zygotes derived from mothers lacking *mbk-2* (hereafter referred to as *mbk-2* zygotes), oocyte granule disassembly and zygote granule assembly proceeded as in wild-type during meiosis (*Figure 1—figure supplement 1*). The zygote granules, however, failed to disassemble in the anterior cytoplasm during zygote polarization (*Figure 1B*).

PPTR-1 is a B′/B56 regulatory subunit of PP2A that is required for P granule maintenance during mitosis (*Gallo et al., 2010*). In *pptr-1* mutant zygotes, disassembly of oocyte granules and assembly of zygotic granules proceeded normally during meiosis. After meiosis, however, granules in the posterior did not grow (*Figure 1B*) and all granules eventually disassembled by the onset of mitosis, as previously reported (*Gallo et al., 2010*). PPTR-1 is 58% identical at the amino acid level to PPTR-2, another predicted B′/B56 regulatory subunit of PP2A (WormBase; [*Harris et al., 2014*]). To determine whether PPTR-2 might also regulate P granule dynamics, we compared zygotes derived from *pptr-1* mothers and *pptr-1 pptr-2* mothers. In *pptr-1 pptr-2*, zygote granule assembly during meiosis was less robust and disassembly began even earlier during pronuclear migration (*Figure 1B* and *Figure 1—figure supplement 1*). *pptr-2* single mutants appeared as wild-type (data not shown). We conclude that *pptr-1* and *pptr-2* contribute redundantly to the assembly/stabilization of P granules in zygotes.

Intriguingly, when we depleted *mbk-2* in *pptr-1 pptr-2*, we observed the same phenotype as in *mbk-2*: new P granules assembled in the zygote and remained stable through the first division (*Figure 1B*). *mbk-2* therefore is epistatic to *pptr-1 pptr-2*, as might be expected in a 'no phosphorylation' scenario. These observations suggest that MBK-2 and PP2A$^{PPTR-1/2}$ work in opposition to promote P granule disassembly (MBK-2) and reassembly (PP2A$^{PPTR-1/2}$). One possibility is that MBK-2 and PP2A$^{PPTR-1/2}$ share substrates that, when phosphorylated, destabilize P granules in zygotes (working model in *Figure 1B*).

## Three intrinsically disordered proteins bind to PPTR-1

To identify potential substrates of PP2A$^{PPTR-1/PPTR-2}$, we screened a LexA-PPTR-1 fusion against cDNAs from a mixed-stage *C. elegans* library fused to the GAL-4 activation domain. From $1.2 \times 10^7$ transformants, we obtained ~4500 positive colonies. We sequenced plasmids from 111 colonies and identified 32 unique genes (*Figure 2—source data 1*). We tested each gene by RNAi (1) for defects in P granule distribution in wild-type embryos, or (2) for suppression of the *pptr-1* hyper-granule disassembly phenotype and identified three genes: *meg-1*, F52D2.4, and C36C9.1 (*Figure 2—source data 1*). *meg-1* (and its paralog *meg-2*) was identified previously as a P granule component specific to early embryos

(*Leacock and Reinke, 2008*). F52D2.4 was identified previously in a yeast two-hybrid screen as a *gex-3* interactor and named *gei-12* (*Tsuboi et al., 2002*). Based on our findings (see below), we have renamed this gene *meg-3*. C36C9.1 is 71% identical to F52D2.4/*meg-3* at the amino acid level, and we have named it *meg-4* (*Figure 2*). *meg-3* and *meg-4* were identified previously as *pptr-2* interactors in a genome-wide yeast two-hybrid screen (*Simonis et al., 2009*). All four *meg* genes are X-linked and code for 70–95.8 kDa proteins that contain 11.8–14.3% serine and 7.2–9.6% asparagine residues and >60% residues predicted to be in disordered regions (*Figure 2*). *meg-1/2,* however, are not homologous to *meg-3/4* (WormBase; [*Harris et al., 2014*]).

To test that the MEG-1, MEG-3, and MEG-4 proteins interact directly with PPTR-1 and PPTR-2, we performed GST pull down assays (*Figure 3A*). We found that all three bound to PPTR-1 and/or PPTR-2 but not to maltose-binding protein (MBP) or to PAA-1, the scaffolding subunit of PP2A. MEG-1, MEG-3, and MEG-4 also interacted with the P granule component PGL-1 (*Figure 3A*).

## MEG-3 and MEG-1 are MBK-2 and PPTR-1 substrates

To test whether the MEGs are also MBK-2 substrates, we expressed MEG-1 and MEG-3 as recombinant MBP fusions in *Escherichia coli* and performed in vitro kinase assays with recombinant MBK-2 as in *Cheng et al. (2009)*. MBK-2 could phosphorylate MBP::MEG-1 and MBP::MEG-3, but not MBP alone (*Figure 3B*). Using the DYRK consensus site K/R $X_{1–3}$ S/T P (*Himpel et al., 2000*; *Campbell and Proud, 2002*), we identified putative consensus sites in MEG-1 (S574) and MEG-3 (T541, S582, T605). S574 in MEG-1 is reported as a phosphorylated site in Phosida (*Gnad et al., 2007*). We mutated each consensus site to alanine and repeated the kinase assays. Phosphorylation was reduced but not eliminated in the alanine mutants (*Figure 3B*). We conclude that recombinant MBK-2 phosphorylates MEG-1 and MEG-3 on consensus sites in vitro, as well as other sites that remain to be determined.

To examine whether MEG-1 and MEG-3 are phosphoproteins in vivo, we ran lysates from embryos expressing GFP::MEG-1 or GFP::MEG-3 on Phos-tag SDS-PAGE. Phos-tag retards the migration of phosphorylated proteins (*Kinoshita et al., 2006*). We detected one primary species for both GFP::MEG-1 and GFP::MEG-3 and several higher molecular weight species that did not resolve well into specific bands. Treatment with alkaline phosphatase eliminated the higher molecular weight species and increased the intensity of the lower species (unphosphorylated isoform, *Figure 3C*, *Figure 3—figure supplement 1*). The relative abundance of the unphosphorylated isoform increased in lysates depleted of *mbk-2* and decreased in lysates depleted of *pptr-1* (*Figure 3C,D*). We conclude that MEG-1 and MEG-3 are phosphoproteins in vivo and that MBK-2 and PPTR-1 promote and antagonize, respectively, their phosphorylation. Under all conditions, GFP::MEG-1 and GFP::MEG-3 still migrated as a mixture of phosphorylated and unphosphorylated isoforms. Because our assays were done in lysates that were depleted for MBK-2 by RNAi, and lysates that lacked PPTR-1 but retained PPTR-2, we do not know whether MBK-2 and PP2A$^{PPTR–1/2}$ are the only regulators of MEG-1 and MEG-3 phosphorylation or whether other kinases and phosphatases are also involved.

## MEG-3 and MEG-4 are required for granule assembly in embryos

To determine the loss of function phenotype of *meg-3*, we obtained a deletion allele of *meg-3* from the National Bioresource Project (*Mitani, 2009*). tm4259 is an out-of-frame, 623-nucleotide deletion that creates a frameshift at amino acid 165 followed by premature stop at amino acid 170 and therefore is a likely null (*Figure 2*). Since RNA-mediated interference experiments suggested that *meg-3* and *meg-4* function redundantly (*Figure 4—figure supplement 1*), we generated a *meg-4* allele in the *meg-3(tm4259)* background by non-homologous end joining repair of CRISPR/Cas9-induced cuts in *meg-4* (*Paix et al., 2014*). *meg-4(ax2026)* is a 7 bp deletion which causes a frameshift at amino acid 13, followed by a premature stop at amino acid 48 (*Figure 2*). We also generated a second *meg-4* allele in the wild-type background by non-homologous end joining repair of two CRISPR/Cas9 cuts flanking the *meg-4* ORF (*Paix et al., 2014*). *meg-4(ax2081)* is a 3.2 kb deletion that removes 733 bases upstream of the *meg-4* ATG and 2565 bases of the *meg-4* coding sequence (total length: 2589 bases).

To determine whether *meg-3* and *meg-4* are required for P granule dynamics, we crossed in GFP::PGL-1 and mCherry::histone transgenes to analyze P granule dynamics in the *meg-3*, *meg-4* single and double mutants (*Figure 4*). We observed the strongest phenotype in the double mutant. First, we noticed that disassembly of oocyte granules was incomplete in *meg-3 meg-4* zygotes (*Video 2*, *Figure 1—figure supplement 1*), causing a few oocyte granules to persist through the first mitotic

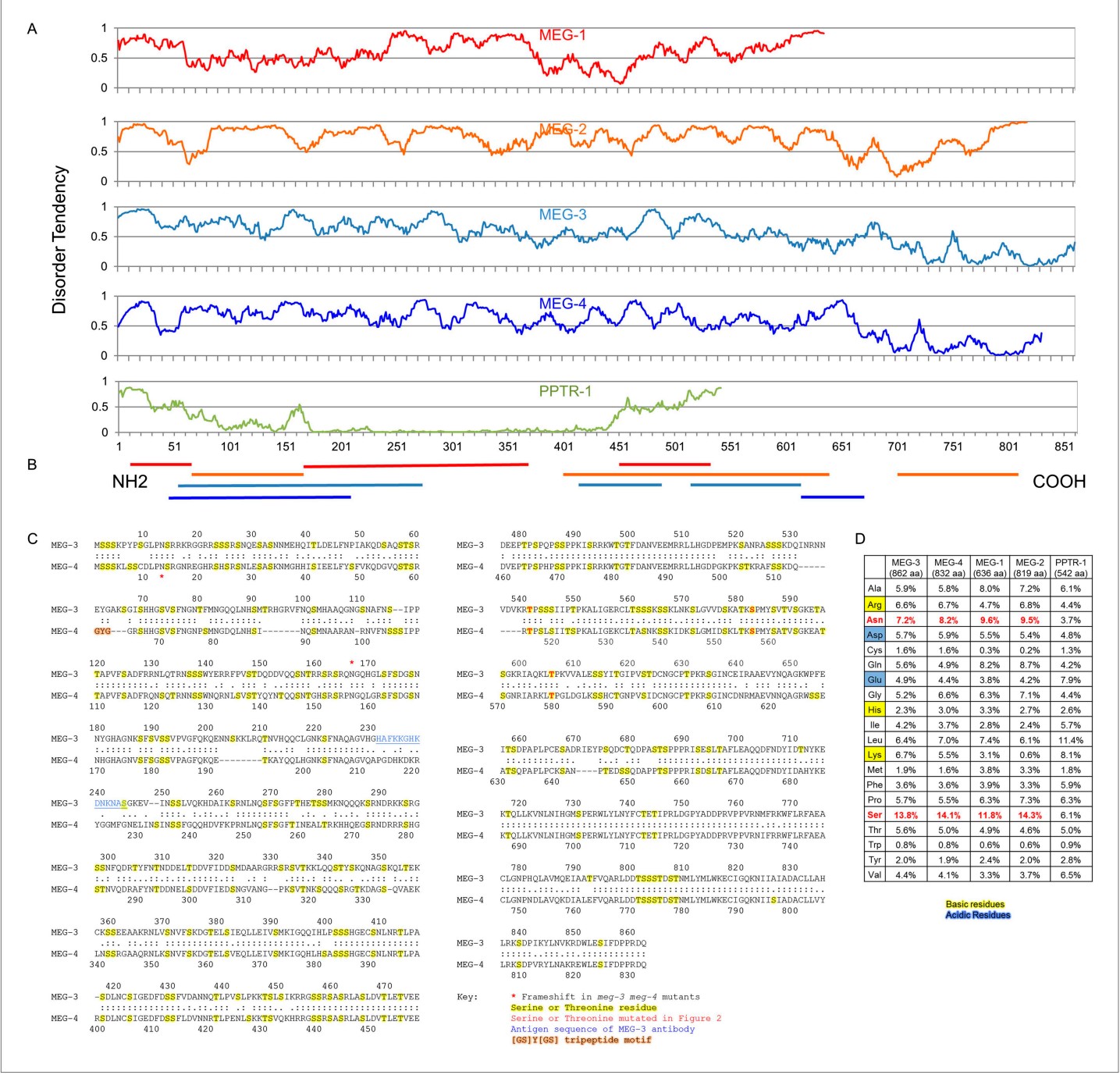

**Figure 2**. The MEG proteins contain sequences predicted to be intrinsically disordered. (**A**) Graphs showing the disorder tendency of sequences along each protein calculated using IUPRED (http://iupred.enzim.hu/) using 'long disorder' parameters (**Dosztányi et al., 2005**). Scores above 0.5 indicate disorder. Number of predicted disordered residues: MEG-1: 420 of 636 aa (66%), MEG-2: 715 of 819 aa (87%), MEG-3: 542 of 862 aa (63%), MEG-4: 570 of 832 aa (69%), PPTR-1: 97 of 542 aa (18%). PPTR-1 is included here as an example of a mostly ordered protein. (**B**) Lines below graph: regions of low complexity sequence as defined using SEG (http://mendel.imp.ac.at/METHODS/seg.server.html) (**Wootton, 1994**). Using the default SEG parameters (SEG 12 2.2 2.5) as used in **Kato et al. (2012)**, there are no low complexity sequences greater than 36 residues in the MEG proteins. Using a larger window size (SEG 45 3.4 3.75), all MEGs have predicted low complexity regions as shown, but PPTR-1 does not. (**C**) Protein sequence alignment of MEG-3 and MEG-4, which are 71% identical to each other. Yellow highlight: serine and threonine residues. Red: predicted MBK-2 phosphorylation site mutated in **Figure 3** kinase assay. Blue: antigen sequence of MEG-3 antibody. Orange highlight: [GS]Y[GS] tripeptide motif (**Kato et al., 2012**). There is only 1 instance of this sequence in MEG-4 and none in MEG-3. Red star: position of the frameshift in the *meg-3(tm4259)* and *meg-4(ax2026)* mutants. *meg-3(tm4259)* is a deletion of 623 nucleotides starting at nucleotide 543 (amino acid 165), followed by an insertion of 'TACGA'. The frameshift inserts

*Figure 2. Continued on next page*

*Figure 2. Continued*

the sequence 'KQGRH' at amino acid 165 followed by a premature stop. *meg-4(ax2026)* is a deletion of 7 nucleotides starting at nucleotide 37 (amino acid 13). The frameshift inserts the amino acids 'EETEKVTALIVEAIWKVHRKIWDTTLVLKNCFIRS' at amino acid 13, followed by a premature stop. See *Table 2* for full allele descriptions. (**D**) Amino acid composition of indicated proteins. Basic residues are highlighted in yellow, acidic in blue. Red residues are overrepresented in MEG proteins.
The following source data is available for figure 2:

**Source data 1**. Candidates from yeast two-hybrid screen.

division (arrows in *Figure 4A*). During mitosis, a few new granules appeared in the posterior cytoplasm (*Figure 4A*), but these remained small (<1 micron) and most were not maintained during the division (*Video 2*). As a result, the total number of granules (including oocyte granules) in *meg-3 meg-4* zygotes in mitosis was ~11% that of wild-type (*Figure 4B*). In the 2-cell stage, new small granules formed again transiently in P$_1$, but again these were fewer and smaller than in wild-type (*Figure 4A*). This pattern was repeated at each division such that by the 28-cell stage, only a few scattered granules were observed throughout the embryo, with no detectable enrichment in the germline founder cell P$_4$ (*Figure 4A*). We observed the same P granule dynamics when staining *meg-3 meg-4* zygotes with an antibody against another P granule component, the Vasa-homolog GLH-2 (*Figure 4—figure supplement 1*). *meg-3 meg-4* zygotes showed similar levels of overall GFP:PGL-1 fluorescence compared to wild-type and similar levels of PGL-3 by western blot analysis (*Figure 4—figure supplement 5*). We conclude that the loss of *meg-3* and *meg-4* causes a dramatic reduction in the number of P granules but does not affect PGL-1 and PGL-3 levels significantly.

*meg-3* single mutants exhibited the same phenotypes as the *meg-3 meg-4* double mutant but with lower expressivity. In *meg-3* zygotes, oocyte granule disassembly was delayed but was eventually completed before the first division (*Video 3*). Zygote granule assembly was also impaired, but less than in *meg-3 meg-4* zygotes: we observed a 30% reduction in the number of P granules in *meg-3* zygotes compared to the 89% reduction for *meg-3 meg-4* zygotes (*Figure 4B*). We also observed only a slight reduction in P granule number in *meg-4* zygotes (*Figure 4B*). We conclude that *meg-3* and *meg-4* contribute redundantly to the assembly of zygote granules, with *meg-3* providing the greater contribution.

We next tested whether RNA components of P granules are affected in *meg-3 meg-4* embryos. In wild-type, during each asymmetric division of the germline P blastomeres, the P granule-associated RNA *nos-2* segregates preferentially with the P granules (asymmetric segregation) (*Subramaniam and Seydoux, 1999*). The lower levels inherited by the somatic daughter are turned over after division (degradation). In *meg-3 meg-4* zygotes, we found that *nos-2* RNA was distributed equally to P blastomeres and their somatic sisters, consistent with the lack of stable P granules. *nos-2* RNA, however, was still quickly degraded in the somatic blastomeres after division as in wild-type (*Figure 4—figure supplement 2A*). Thus, as previously reported for *pptr-1* mutants (*Gallo et al., 2010*), despite symmetric segregation of *nos-2* RNA, an inherent asymmetry between somatic and germline blastomeres in *meg-3 meg-4* embryos causes *nos-2* RNA to be maintained only in the germline blastomeres. Loss of *meg-3* and *meg-4* also did not affect the formation of P bodies (*Figure 4—figure supplement 2B*) nor the distribution of asymmetric proteins that segregate independently of P granules (PIE-1 and MEX-5) (*Figure 4—figure supplement 2C*). By the L1 larval stage, when new P granule components are synthesized in the PGCs (*Kawasaki et al., 2004*), low levels of perinuclear P granules became visible in the primordial germ cells of *meg-3 meg-4* double mutants (*Figure 4D*). By the L4 larval stage, P granule levels were indistinguishable from wild-type in fertile *meg-3 meg-4* mutants (*Figure 4—figure supplement 2D*). We conclude that MEG-3 and MEG-4 are only required for P granule assembly in pre-gastrulation embryos and are not required for other soma/germline asymmetries, or for the assembly of P bodies or perinuclear P granules.

## MEG-1 contributes to granule assembly and disassembly and is required redundantly with MEG-3 and MEG-4 for fertility

To determine the role of *meg-1* and *meg-2* in P granule assembly and disassembly, we again used the GFP::PGL-1 and mCherry::H2B transgenes to follow granule dynamics live. *Leacock and Reinke (2008)*

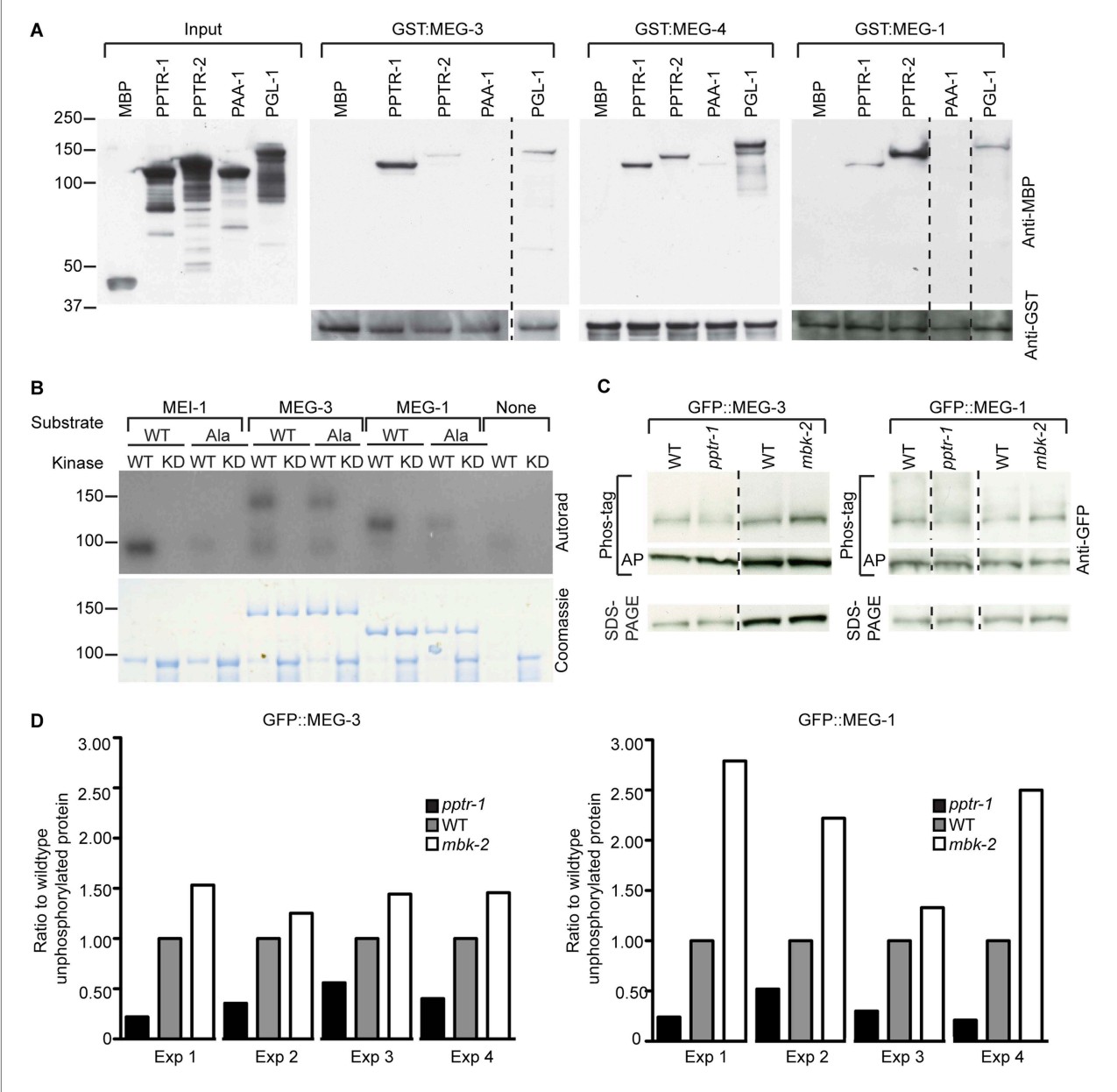

**Figure 3**. MEG-1 and MEG-3 are substrates of MBK-2 and PP2A[PPTR–1/2]. (**A**) Westerns of *E. coli* lysates expressing the indicated MBP fusions before (input) and after immobilization on columns containing the indicated GST fusions. MBP and PAA-1 are negative controls. PAA-1 is the scaffolding subunit of PP2A. Dashed lines separate individual gels. (**B**) In vitro kinase assay. MBP fused to wild-type or kinase-dead (KD) MBK-2 was incubated with gamma[32] P-ATP and MBP-fused substrates: MEI-1 (94 kDa); MEI-1(S92A) (94 kDa); MEG-3 (138 kDa with degradation band at ~100 kDa); MEG-3(T541A S582A T605A) (138 kDa with degradation band at ~100 kDa); MEG-1 (112 kDa); MEG-1(S574A) (112 kDa). MBP::MBK-2 is 99 kDa and autophosphorylates. MEI-1 is a previously known substrate of MBK-2 (*Stitzel et al., 2006*). Coomassie staining to control for loading is shown below. Phosphorylation is diminished in MEG-3(T541A S582A T605A) (64% of wild-type phosphorylation) and MEG-1 (S574A) (88% of wild-type phosphorylation). (**C**) Anti-GFP westerns of *C. elegans* embryonic lysates run on SDS-PAGE gels with (top) or without Phos-tag (bottom). Wild-type, *pptr-1(tm3103)*, and *mbk-2(RNAi)* embryonic lysates expressing GFP::MEG-3 or wild-type, *pptr-1(RNAi)*, and *mbk-2(RNAi)* embryonic lysates expressing GFP::MEG-1 were treated with or without alkaline phosphatase (AP) and equal amounts were loaded on gels with or without Phos-tag. Dashed lines separate individual gels. Loading control was protein run on SDS-PAGE without Phos-tag. (**D**) Graphs showing % unphosphorylated protein relative to wild-type in four Phos-tag experimental replicates. % unphosphorylated was calculated as the ratio of the intensity of the band corresponding to unphosphorylated protein on a Phos-tag gel over the intensity of the band on a non-Phos-tag gel (total protein) and normalized to wild-type.

The following figure supplement is available for figure 3:

**Figure supplement 1**. MEG-1 and MEG-3 are phosphorylated in vivo.

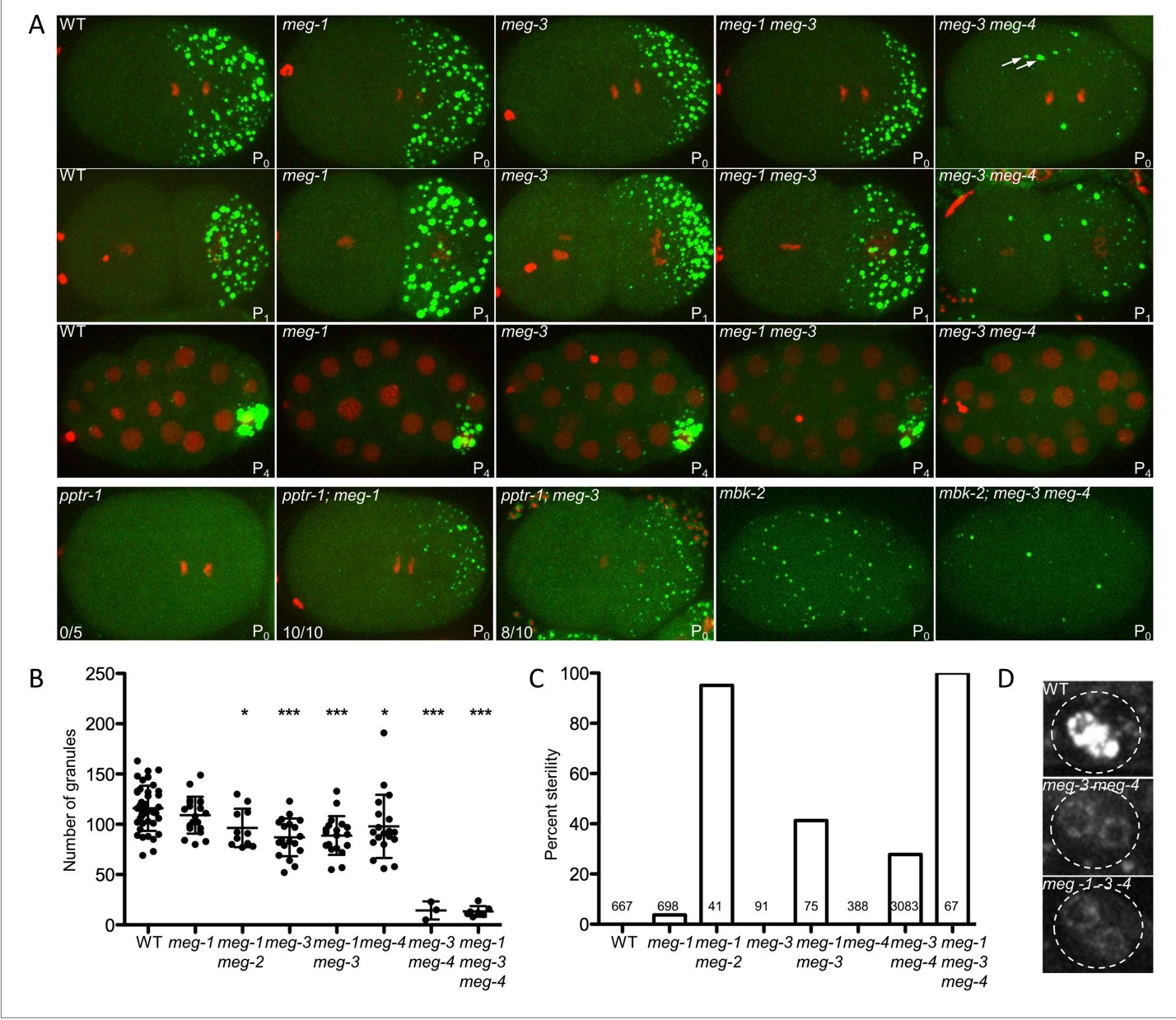

**Figure 4**. *meg-1*, *meg-3*, and *meg-4* regulate P granule dynamics. (**A**) Live embryos of the indicated genotypes and stages expressing GFP::PGL-1 (green) and mCherry::H2B (red) or GFP::PGL-1 only (last two). Arrows in *meg-3 meg-4* point to oocyte granules. For *pptr-1* zygotes, the number of zygotes with visible posterior P granules is indicated. $P_0$ is 1-cell stage, $P_1$ is 2-cell stage, $P_4$ is 28-cell stage. (**B**) Number of GFP::PGL-1 granules in zygotes in anaphase of the first mitosis. Each dot represents a different zygote, and mean and standard deviation are shown. Asterisks indicate data that are statistically significantly different from wild-type (one asterisk: $p < 0.005$, three asterisks: $p < 0.0001$). (**C**) Percent sterility of adult hermaphrodites of the indicated genotypes grown at 20°C. Number of hermaphrodites scored (n) is written above the x axis. (**D**) Primordial germ cells in the L1 larval stage expressing GFP::PGL-1. By this stage, PGCs express P granule components (*Kawasaki et al., 2004*). Perinuclear P granules form in all three genotypes. GFP::PGL-levels are lower in the *meg* mutants due to the lack of P granules inherited from the early embryonic stages. Full genotypes are *meg-1(vr10)*, *meg-1(vr10) meg-2(RNAi)*, *meg-3(tm4259)*, *meg-1(vr10) meg-3(tm4259)*, *meg-4(ax2081)*, *meg-3(tm4259) meg-4(ax2026)*, *pptr-1(tm3103)*, *pptr-1(tm3103); meg-1(vr10)*, *pptr-1(tm3103);meg-3(tm4259)*, *mbk-2(pk1427)* and *mbk-2(pk1427); meg-3(RNAi) meg-4(RNAi)*.

The following figure supplements are available for figure 4:

**Figure supplement 1**. *meg-3* and *meg-4* are required redundantly for P granule assembly in zygote.

**Figure supplement 2**. Comparison of wild-type and *meg-3 meg-4* mutants.

*Figure 4. Continued on next page*

*Figure 4. Continued*

**Figure supplement 3**. *meg-1 meg-2* embryos exhibit defects in P granule disassembly in 2-cell and later embryos leading to missegregation of P granules to somatic blastomeres.

**Figure supplement 4**. MEG proteins are required for germ cell development.

**Figure supplement 5**. *meg-3 meg-4* do not affect PGL-1 and PGL-3 levels significantly.

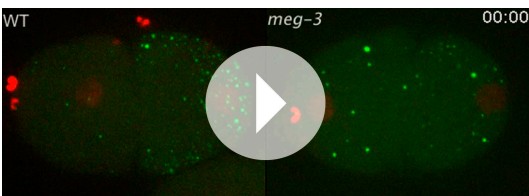

**Video 3**. P granule dynamics in wild-type and meg-3. Time-lapse of embryos expressing GFP::PGL-1 and mCherry::H2B. Wild-type embryo is same one as shown in **Video 2**. Images are maximum intensity projections of 8 z planes separated by 1 μm steps. Stacks were taken every 8 s, total movie time is 17 min 20 s, movie is played back in 80× real time. Full genotype of mutant is *meg-3(tm4259)*. The bright red puncta in the anterior (left) side of the embryos are polar bodies. In the wild-type movie, there are also red puncta above the embryo, these are sperm outside of the egg shell.

reported that *meg-1* and *meg-1 meg-2* mutants assemble P granules, but occasionally missegregate P granules to somatic blastomeres in 4-cell and later stage embryos. Consistent with those findings, we observed only a modest reduction in the number of P granules in *meg-1 meg-2* zygotes (***Figure 4A,B***). In 2-cell stage *meg-1* mutants, P granules failed to disassemble in the anterior cytoplasm of the $P_1$ blastomere (***Figure 4A***), causing a few P granules to be inherited by the EMS blastomere (***Figure 4—figure supplement 3A***). *meg-2(RNAi)* on *meg-1* mutants enhanced the percent of P granule missegregation to somatic blastomeres (***Figure 4—figure supplement 3B***), as reported previously (***Leacock and Reinke, 2008***). We conclude that, unlike *meg-3* and *meg-4*, *meg-1* and *meg-2* contribute only weakly to granule assembly and contribute primarily to granule disassembly.

To determine whether the *meg* genes act redundantly in granule assembly, we compared double, triple, and quadruple loss-of-function zygotes, combining mutations and RNAi (see ***Figure 4*** legend for complete genotypes). We found that the P granule assembly defects observed in *meg-3* and *meg-3 meg-4* zygotes were not affected by the additional loss of *meg-1* or *meg-1* and *meg-2* (***Figure 4A*** and ***Figure 4—figure supplement 3C***). In the L1 stage, we observed low levels of perinuclear P granules in the PGCs of *meg-1 meg-3 meg-4* triple mutants as seen in *meg-3 meg-4* double mutants (***Figure 4D***). We conclude that *meg-3* and *meg-4* are the primary contributors to granule assembly in early embryos, and that none of the *meg*s are required for the assembly of perinuclear granules in PGCs later in development.

*meg-1 meg-2* double mutants are 100% sterile (***Leacock and Reinke, 2008***). To examine whether other *meg* mutant combinations are also sterile, we determined the percent of adult animals with empty uteri (sterile animals). We found that 27% of *meg-3 meg-4* mutants and 100% of *meg-1 meg-3 meg-4* mutants were sterile (***Figure 4C***). In wild-type animals, the two PGCs begin to divide in the first larval stage (L1) to generate >1000 germ cells by adulthood. In contrast, we observed <10 germ cells in *meg-1 meg-3 meg-4* larvae and no further growth by the adult stage, despite the presence of a somatic gonad (***Figure 4—figure supplement 4***). This phenotype is identical to that reported for *meg-1 meg-2* mutants (***Leacock and Reinke, 2008***). We conclude that *meg-1*, *meg-2*, *meg-3*, and *meg-4* contribute redundantly to germ cell proliferation during larval development.

## The MEGs function downstream of MBK-2 and PPTR-1 to regulate P granule dynamics

To test whether the *meg* genes are epistatic to *pptr-1* and *mbk-2*, we examined the effect of loss of *meg-1* and *meg-3* in *pptr-1* and *mbk-2* zygotes. In *pptr-1* mutants, all P granules disassemble during mitosis. If this defect were due to hyper-phosphorylation of a PPTR-1 target, then elimination of that

target might suppress the *pptr-1* phenotype and restore some P granules. Consistent with this possibility, we found that the loss of *meg-1* suppressed the granule hyper-disassembly phenotype of *pptr-1* zygotes (*Figure 4A*). Interestingly, the loss of *meg-3* also partially suppressed *pptr-1* (*Figure 4A*), indicating that *meg-3* also contributes to disassembly.

In *mbk-2* mutants, P granule disassembly does not occur during mitosis. We found that in *mbk-2, meg-3 meg-4* zygotes, P granules failed to assemble as is observed in *meg-3 meg-4* zygotes (*Figure 4A*). This finding indicates that *meg-3* and *meg-4* are essential for granule assembly even when disassembly is inhibited. Together with the finding that *mbk-2* is epistatic to *pptr-1/2* (*Figure 1*), these genetic interactions are consistent with the *meg* genes functioning downstream of *mbk-2* and *pptr-1/2*, to regulate the balance of granule assembly and disassembly in embryos.

## MEG-3 and MEG-4 localize to embryonic P granules

The experiments above suggest a direct role for the MEGs in disassembling and assembling P granules. MEG-1 was previously reported to localize to embryonic P granules from the 4-cell stage to the 100-cell stage (*Leacock and Reinke, 2008*). To determine the localization of MEG-3, we generated a rescuing GFP-tagged transgene and a polyclonal serum raised against a MEG-3 peptide ('Materials and methods' and *Figure 5—figure supplement 1*). For MEG-4, we inserted a 3× FLAG tag at the carboxy terminus of the *meg-4* open reading frame by homology dependent repair of a CRISPR/Cas9-induced cut (*Paix et al., 2014*). We found that both MEG-3 and MEG-4 localize to P granules from the 1-cell stage to ~100-cell stage (*Figure 5* and *Figure 5—figure supplement 2A*). After the birth of the primordial germ cells Z2/Z3 (100-cell stage), when P granules are fully perinuclear and no longer dynamic, MEG-3 and MEG-4 levels quickly faded (*Figure 5*, *Figure 5—figure supplement 2* and data not shown).

We did not detect MEG-3 or MEG-4 in the perinuclear P granules of adult gonads (*Figure 5A* for MEG-3 and data not shown). We detected GFP::MEG-3 in the cytoplasm of immature germ cells and oocytes (*Figure 5A*). In zygotes, cytoplasmic GFP::MEG-3 formed an anterior-to-posterior gradient with highest levels in the posterior (*Figure 5C*). In embryos depleted for *pgl-1* and *pgl-3*, which do not form P granules (*Hanazawa et al., 2011*), GFP::MEG-3 was still present in an anterior-to-posterior gradient but no longer localized to large granules (*Figure 5C*). We conclude that MEG-3 and MEG-4 are maternally provided proteins that segregate with the P lineage and associate with P granules during the embryonic period where P granules are most dynamic.

Live-imaging of zygotes expressing GFP::MEG-3 and mCherry::PGL-3 revealed that, during granule disassembly, GFP::MEG-3 perdures longer in the granules than mCherry::PGL-3 (*Video 4* and *Figure 5—figure supplement 2C*). Consistent with that observation, in fixed zygotes, we observed MEG-4-positive/PGL-1-negative and MEG-3-positive/PGL-1-negative granules at the anterior most edge of the P granule domain (*Figure 5B* and *Figure 5—figure supplement 2B*). We also observed GFP::MEG-3-positive/PGL-1-negative granules in *pptr-1* zygotes where disassembly is enhanced (*Figure 5—figure supplement 2D*). We conclude that MEG-3/4 and PGL-1 exhibit different dynamics during disassembly.

## MEG-3 localizes to a highly dynamic peri-granular domain

Consistent with the different MEG and PGL dynamics, immunostaining experiments in fixed embryos suggested that MEG-1 and MEG-3 do not co-localize precisely with PGL-1 in P granules (*Figure 5—figure supplement 3A* and *Figure 5—figure supplement 4*). To avoid potential artifacts from fixation, we turned to lattice light sheet microscopy, a new method that uses ultra-thin light sheets derived from optical lattices to image 3D volumes with high temporal and spatial resolution (*Chen et al., 2014*). Imaging of live zygotes co-expressing GFP::MEG-3 and mCherry:PGL-3 revealed that both were non-uniformly distributed and constantly rearranged within the granules (*Figure 5D*, *Video 5*). The GFP::MEG-3 and mCherry::PGL-3 signals overlapped but were never perfectly coincident, even during granule fusion (*Figure 5D*, *Figure 5—figure supplement 3B*). In 34 of 37 granules analyzed, the GFP::MEG-3 signal extended over a larger area than the mCherry::PGL-3 signal (*Figure 5—figure supplement 3C*), with GFP::MEG-3 extending further out at the periphery of each granule compared to mCherry::PGL-3.

Images acquired every 580 ms revealed changes in the distribution of GFP::MEG-3 and GFP::PGL-3 at every time point (*Figure 5—figure supplement 3B*). Overall granule shape was also dynamic, with none of the granules maintaining a perfectly spherical shape (*Figure 5D*, *Video 5*). Fusion between

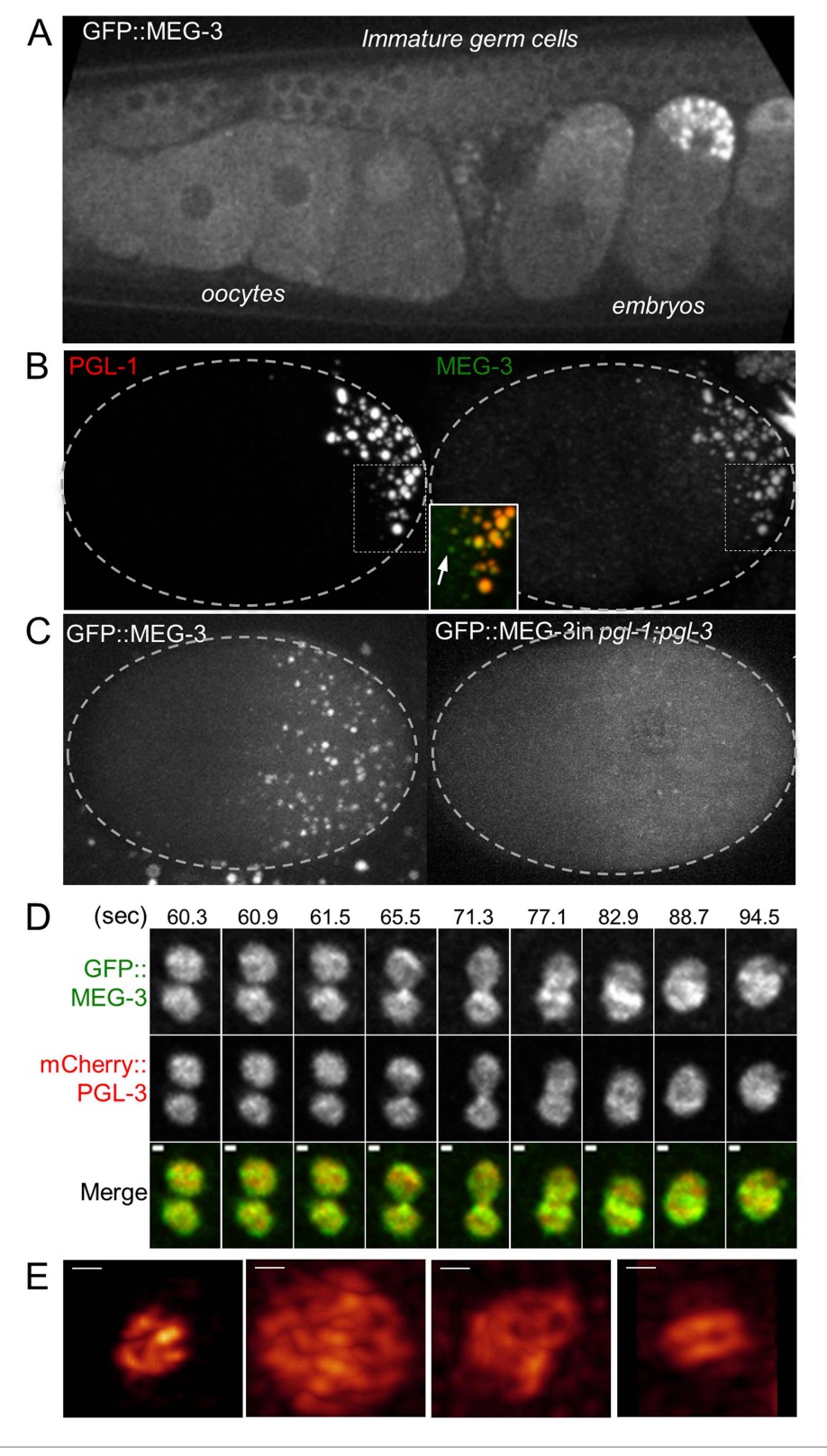

**Figure 5**. MEG-3 localizes to a dynamic domain that surrounds and penetrates the P granules. (**A**) Gonad of an adult hermaphrodite expressing the GFP::MEG-3 transgene under the control of the *meg-3* promoter and 3′ UTR. GFP::MEG-3 associates strongly with P granules only in embryos. (**B**) Fixed wild-type zygote stained with
*Figure 5. Continued on next page*

*Figure 5. Continued*

anti:MEG-3 and anti-PGL-1 sera. Inset: magnification of merged image. Arrow points to a MEG-3-positive/ PGL-1-negative granule. (**C**) Live wild-type and *pgl-1(RNAi);pgl-3(bn104)* zygotes expressing GFP::MEG-3. GFP::MEG-3 localizes to a cytoplasmic gradient and P granules. P granule localization is lost in *pgl-1(RNAi); pgl-3(bn104)* zygotes, but the cytoplasmic gradient remains. (**D**) Still images from a movie acquired using lattice light sheet microscopy (*Video 5*). Max intensity projection of a Z stack through a pair of fusing granules. Time in seconds (*Video 5*) is indicated above each panel. GFP::MEG-3 and mCherry:PGL-3 domains are not completely co-localized. Also see *Figure 5—figure supplement 3B*. Resolution is 238 nm × 238 nm × 500 nm. Scale bars: upper left = 500 nm. (**E**) Lattice light sheet 3D-SIM mode reconstruction of GFP::MEG-3 in P granules in a living zygote (also see *Video 6*). Scale bars: upper left = 500 nm. First granule on the left: acquisition time is 1 s (*Video 6*). Subsequent granules: acquisition time is 1.7 s (*Video 7*).

The following figure supplements are available for figure 5:

**Figure supplement 1**. GFP::MEG-3 rescues *meg-3* mutant and the MEG-3 antibody is specific.

**Figure supplement 2**. MEG-4 localizes to P granules and MEG-3 and MEG-4 assemblies persist longer in disassembling granules than PGL-1.

**Figure supplement 3**. MEG-3 and PGL-1/3 overlap only partially in P granules.

**Figure supplement 4**. MEG-1 only partially co-localizes with PGL-1.

**Figure supplement 5**. Lattice light sheet microscopy identifies sub-granular MEG-3 domain.

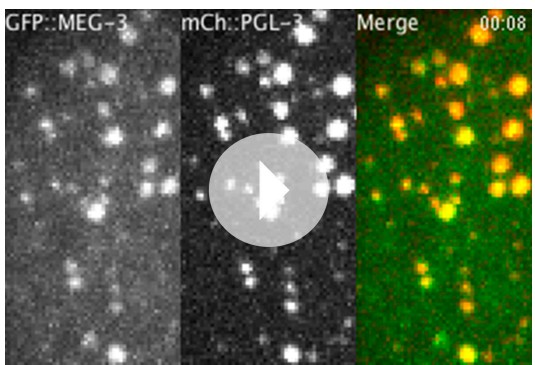

**Video 4**. MEG-3 and PGL-3 dynamics in the anterior of $P_1$ cell. Time-lapse of embryos expressing GFP::MEG-3 and mCherry::PGL-3. Images are maximum intensity projections of 8 z planes separated by 1 μm steps. Stacks were taken every 8 s, total movie time is 9 min 12 s, movie is played back in 80× real time.

granules was comparatively slow, on the order of tens of seconds as documented previously (*Brangwynne et al., 2009*). Granules remained close to each other for several seconds before initiating fusion. In the typical example shown in *Figure 5D* and *Video 5*, the fusing granules took 25 s to return to a quasi-spherical shape. We also detected smaller, dynamic GFP::MEG-3 assemblies in the cytoplasm away from the granules (*Video 5*, *Figure 5—figure supplement 5*).

To examine the distribution of GFP::MEG-3 at higher resolution, we collected single-color, single time point light sheet images in the structured illumination (SIM) mode. In this mode, individual images are collected as the lattice light sheet is stepped along the x axis and reconstructed into a 3D image with resolution beyond the diffraction limit in the x and z directions (194 nm × 238 nm in xz; [*Chen et al., 2014*]). The single time point SIM images confirmed that GFP::MEG-3 is not uniformly distributed in the granules. The non-uniform pattern of GFP::MEG-3 was also visible in raw, unprocessed images (*Figure 5—figure supplement 5*). In 5 out of 5 granules reconstructed in SIM mode, GFP::MEG-3 localized to a discontinuous ribbon-like domain that surrounded and penetrated the granule (*Figure 5E* and *Videos 6 and 7*). We conclude that P granules are not homogeneous assemblies and that MEG-3 localizes to dynamic sub-domains within each granule that only partially overlap with PGL-3.

## Discussion

In this study, we identify the substrates of MBK-2/DYRK that regulate the condensation and dissolution of P granules in *C. elegans* embryos. Our findings indicate that the MEGs are required to stabilize the

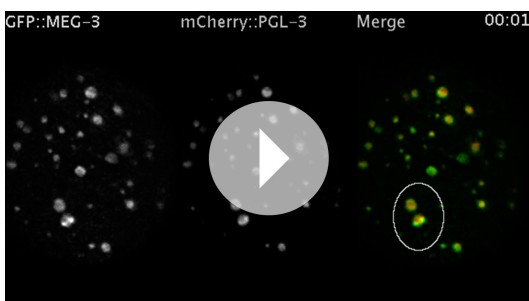

**Video 5**. Lattice light sheet movie of an embryo expressing GFP::MEG-3 and mCherry::PGL-3. Time-lapse of the posterior cytoplasm of a zygote (anterior to the left, posterior to the right) acquired using lattice light sheet microscopy in dithered mode. Two fusing granules are highlighted (also shown in **Figure 5D**). Images are maximum intensity projections of 11 z planes separated by 0.3 µm steps, capturing the entire depth of the fusing pair. Stacks were taken every 580 ms, total movie time is 34 s, movie is played back in 5× real time.

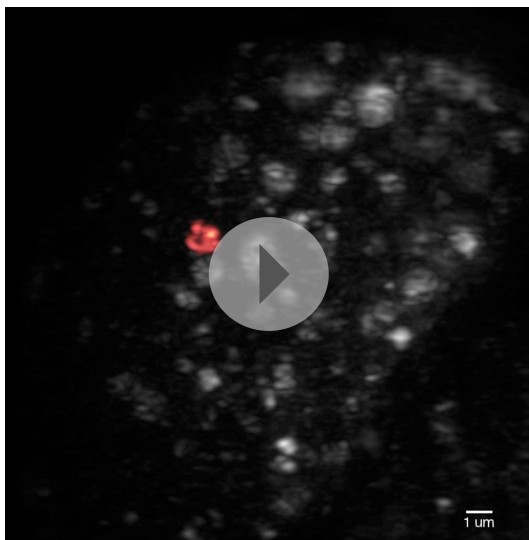

**Video 6**. 3D reconstruction of GFP::MEG-3 granule. Single time point SI images acquired with lattice light sheet microscopy of an individual P granule in an embryo expressing GFP::MEG-3. Acquisition time was 1 s. Resolution is 194 nm × 238 nm × 419 nm. Anterior is to the right, posterior to the left.

condensed phase of P granules. Phosphorylation of the MEGs by the anteriorly enriched kinase MBK-2/DYRK and dephosphorylation by PP2A$^{PPTR-1/PPTR2}$ antagonizes and promotes, respectively, the stabilizing function of the MEGs. The MEGs also contribute redundantly to a second activity required for germline proliferation during larval development. We discuss each aspect of this model below.

## The MEGs stabilize P granules in embryos: possible scaffolding function

During the oocyte-to-zygote transition, oocyte granules disassemble and are replaced by smaller and much more dynamic granules (zygote granules). We suggest that this transition depends on scaffolding of new granules by MEG assemblies throughout the cytoplasm. We propose that, shortly after fertilization, MEG assemblies promote PGL condensation, depleting the cytoplasmic PGL-1 pool available to maintain oocyte granules and causing PGL-1 to redistribute from oocyte granules to MEG-scaffolded zygote granules. In *meg* mutants, this competition does not take place, allowing oocyte granules to persist longer. The alternative possibility that the primary role of the MEGs is to disassemble oocyte granules, which indirectly is required for zygote granule assembly, is unlikely since (1) the *meg-3* single mutant disassembles oocyte granules, yet is still partially defective in zygote granule assembly (**Figure 4**), (2) *mbk-2*, which is required for the disassembly of zygote granules, is not required for the disassembly of oocyte granules (**Figure 1—figure supplement 1**), and (3) oocyte granules represent less than 1% of the total PGL-1 pool available in *meg-3 meg-4* zygotes, so their mere presence is unlikely to interfere with the assembly of new granules (**Figure 4—figure supplement 5**). We detected GFP::MEG-3 in the cytoplasm of oocytes before fertilization, yet loss of *meg-3* does not affect oocyte granules until after fertilization. We do not know what cue activates the scaffolding function of MEG proteins specifically in zygotes.

How do MEG proteins stimulate PGL-1 condensation? Unlike the RNA-binding proteins that associate with hydrogels in vitro (**Frey et al., 2006**; **Kato et al., 2012**), the MEGs do not contain a recognizable RNA-binding domain or short repeated motifs (**Figure 2**). The MEGs contain, however, extended regions with predicted high disorder (**Figure 2**). Intrinsically disordered proteins have been shown to oligomerize and organize scaffolds that template supramolecular structures (**Turoverov et al., 2010**; **Toretsky and Wright, 2014**). For example, ameloblastin self-assembles in vitro into ribbons that can extend to hundreds of nanometers in length (**Wald et al., 2013**). In vivo, ameloblastin is essential for the organization of enamel matrix proteins during teeth biomineralization. MEG-1 and MEG-3 bind directly to PGL-1 in vitro, consistent

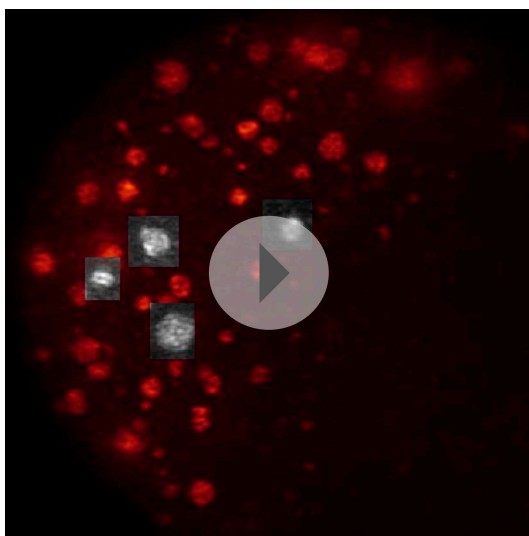

**Video 7**. 3D reconstructions of GFP::MEG-3 granules. Single time point SI images acquired with lattice light sheet microscopy of individual P granules in an embryo expressing GFP::MEG-3. Four examples are shown. Acquisition time was 1.75 s. Anterior is to the right, posterior to the left.

with a role in organizing PGL-1 assemblies. When over-expressed in mammalian cells, PGL-1 can assemble granules on its own (*Hanazawa et al., 2011*). In *meg-3 meg-4* zygotes, a few small PGL-1 granules form transiently in the posterior cytoplasm but are not maintained. We suggest that, in zygotes, the self-assembly properties of PGL-1 are only sufficient to form small, unstable granules, which must be stabilized by a MEG-dependent scaffold. Our observations using light sheet microscopy indicate that MEG-3 localizes to small assemblies in the cytoplasm and larger assemblies surrounding a PGL core in the P granules. During granule disassembly, MEG-3 and MEG-4 assemblies persist longer than the PGL-1 core, consistent with occupying a distinct domain. Together these observations suggest the following working model: the MEGs promote granule assembly by forming dynamic scaffolds in the cytoplasm that stabilize and wrap around PGL condensates.

By electron microscopy, embryonic P granules were reported to appear as homogenous, fibrillar granular bodies (*Wolf et al., 1983*). Our observations suggest that embryonic P granules in fact contain distinct surface and internal zones, as also reported for post-embryonic, perinuclear P granules (*Pitt et al., 2000*; *Sheth et al., 2010*). Nucleoli, which also behave like liquid droplets, also contain a distinct surface shell (*Brangwynne et al., 2011*). Nucleoli, however, are perfect spheres (*Brangwynne et al., 2011*), whereas P granules are imperfect spheres with irregular, asymmetric contours that are constantly in flux. We suggest that this difference is due to the fact that P granules are undergoing continuous exchange with smaller assemblies in the surrounding cytoplasm. Consistent with this possibility, photobleaching experiments have demonstrated rapid exchange of GFP::PGL-1 and GFP::MEG-3 between granules and cytoplasm (*Brangwynne et al., 2009* and data not shown). We conclude that P granules in embryos are not stable, homogeneous droplets, but dynamic assemblies of two (or more) condensates that mix only partially and are constantly in flux with smaller assemblies in the cytoplasm.

## Phosphorylation of the MEGs by MBK-2/DYRK promotes granule disassembly

MEG-1 and MEG-3 are direct targets of MBK-2/DYRK and the PP2A phosphatase. The MEGs are rich in serines (75 serines in MEG-1 and 119 serines in MEG-3), and their migration on Phos-tag gels suggests a complex pattern of phosphorylation with many phosphoisoforms. Initial attempts to mutate individual serines in MEG-1 and MEG-3 have not yielded detectable phenotypes, suggesting that the serines may function cumulatively (data not shown). Accumulation of negatively charged phosphates could interfere with the interactions that stabilize the MEG scaffold or its interactions with PGL-1. Consistent with this possibility, phosphorylation of the low complexity domain of the FUS stress granule protein interferes with its incorporation into hydrogels in vitro (*Han et al., 2012*). Our genetic analyses suggest that all MEGs contribute to assembly and disassembly but to different extents, with MEG-3 and MEG-4 together making the most important contribution to assembly. The predicted pI (isoelectric point) of MEG-1 and MEG-2 in the unphosphorylated state is 6.63 and 6.04, compared to 9.74 and 9.33 for MEG-3 and MEG-4. We speculate that the intrinsic negative charge of MEG-1 and MEG-2, which is increased further by phosphorylation, favors a role for these proteins in disassembly. In contrast, the intrinsic weak positive charge of MEG-3 and MEG-4 allows them to contribute robustly to both disassembly and assembly, as they switch to negative and back to positive upon phosphorylation and dephosphorylation.

How are disassembly and assembly spatially segregated? At the onset of disassembly, MBK-2/DYRK is enriched in the anterior half of the zygote (*Pellettieri et al., 2003*). If dephosphorylation by

PP2A continuously antagonizes phosphorylation by MBK-2, even small changes in the level of MBK-2 levels along the anterior/posterior axis could bias MEG phosphorylation and P granule disassembly. Paradoxically, MBK-2/DYRK also accumulates in P granules, especially late in the first cell cycle as the granules continue to grow in the posterior cytoplasm (*Pellettieri et al., 2003*). Interestingly, DYRK3, the vertebrate homolog of MBK-2 that regulates stress granule dynamics, also localizes in the granules (*Wippich et al., 2013*). Drug inhibition studies suggest that DYRK3 enhances granule condensation when its kinase activity is inhibited and promotes granule dissolution when activated (*Wippich et al., 2013*). An attractive possibility is that, as P granules increase in size, MBK-2 activity decreases within the granule, perhaps due to the accumulation of an inhibitor. Interestingly, GFP::PPTR-1 also becomes enriched in P granules in zygotes (data not shown). There are also other asymmetries in the zygote cytoplasm that could bias P granule dynamics. GFP::MEG-3 forms an anterior-low/posterior-high gradient along the length of the zygote, and the RNA-binding protein MEX-5, which is required for full disassembly, forms an opposing gradient with higher levels in the anterior (*Schubert et al., 2000*). Modeling studies have already demonstrated that, in principle, even weak gradients of phase transition regulators are sufficient to segregate P granules (*Lee et al., 2013*). It will be important to determine which asymmetry in the zygote cytoplasm is directly responsible for patterning P granule dynamics.

## MEG proteins contribute to an essential germ plasm activity that does not require P granules

Analysis of single, double, and triple *meg* combinations indicate that the *meg*s display synthetic sterility. For example, *meg-1* and *meg-3 meg-4* mutants are ~4% and ~30% sterile, respectively, whereas *meg-1 meg-3 meg-4* mutants are 100% sterile (this work and *Leacock and Reinke, 2008*). In the sterile *meg-1 meg-3 meg-4* worms, the primordial germ cells stop dividing soon after initiating divisions in the gonad, a phenotype previously reported for *meg-1 meg-2* mutants (Meg phenotype: **m**aternal-**e**ffect **g**erm cell defective) (*Leacock and Reinke, 2008*). We conclude that the MEGs contribute redundantly to an activity essential for germ cell viability and/or proliferation.

The MEGs are components of the *C. elegans* germ plasm, the specialized maternal cytoplasm that segregates with the embryonic germline and specifies germ cell fate. Germ plasm is thought to have evolved independently several times during metazoan evolution (*Extavour and Akam, 2003*), and the molecules that initiate germ plasm and/or germ granule assembly in eggs are not conserved between different species, although some contain intrinsically disordered regions (e.g., Bucky Ball/Xvelo in vertebrates, MEGs in *C. elegans*) (*Marlow and Mullins, 2008*; *Bontems et al., 2009*; *Nijjar and Woodland, 2013*). In all species examined, the germ plasm contains microscopically visible P granule-like structures that are enriched for conserved mRNAs and RNA-binding proteins required for germ cell development (e.g., *nanos* RNA and VASA-related helicases) (*Voronina et al., 2011*). Despite this conservation, our genetic analyses suggest that the MEGs' contribution to fertility is not linked to their effects on P granules, as also suggested by *Leacock and Reinke (2008)*. *meg-1 meg-2* embryos assemble embryonic P granules and *meg-1 meg-3 meg-4* embryos do not, yet both display the same fully penetrant Meg sterility phenotype (*Figure 4* and *Leacock and Reinke, 2008*). Furthermore, *meg-3 meg-4* embryos show the same dramatic defect in P granule assembly as *meg-1 meg-3 meg-4 embryos*, yet most *meg-3 meg-4* animals (70%) are fertile (*Figure 4*). We conclude that P granule assembly in the germ plasm of early embryos is neither required nor sufficient for fertility in *C. elegans*. In contrast, the perinuclear P granules that form in primordial germ cells and their descendents are required for fertility (*Updike and Strome, 2010*). Consistent with that view, fertile *meg* mutants that do not assemble P granules in the germ plasm assemble perinuclear P granules de novo later in development. De novo assembly of perinuclear germ granules is also observed in animals that do not inherit germ plasm and specify their primordial germ cells through inductive mechanisms later in development (*Voronina et al., 2011*).

Previously, our analyses of *pptr-1* mutants led us to conclude that asymmetric segregation of core P granule components is not essential to distinguish germline from soma in embryos (*Gallo et al., 2010*). Our results here support this view and suggest that the essential activity of the germ plasm resides not in the assembly of P granules per se but in the MEG proteins that regulate granule assembly. How the maternally provided MEGs contribute to the health and proliferating potential of the PGCs will be an interesting area for future exploration.

## Materials and methods

### Worm handling, transgenics, CRISPR-mediated editing, RNAi, sterility counts

*C. elegans* was cultured according to standard methods (*Brenner, 1974*). Transgene plasmids were generated by InFusion cloning (Clontech, Mountain View, CA) and introduced into *unc-119* worms by microparticle bombardment (*Praitis et al., 2001*). GFP::MEG-3 is driven by its endogenous promoter and 3'UTR and consist of 1128 bp of genomic DNA sequence upstream of the MEG-3 ATG (up to but not including the ATG of the next gene Y40A1A.2), GFP from pCM1.53 (*Merritt et al., 2008*) and 1872 bp of genomic DNA downstream of the stop codon, past both polyA sites annotated in WormBase (WBsf257432 and WBsf216670). GFP::MEG-1 is described in *Leacock and Reinke (2008)* and is driven by the *pie-1* promoter and *pie-1* 3'UTR. CRISPR/Cas9 experiments to generate *meg-3* and *meg-4* alleles were performed as described in *Paix et al. (2014)*. See *Tables 1 and 2* for list of strains, allele descriptions, and sgRNA and repair templates sequences.

RNAi was performed by feeding (*Timmons and Fire, 1998*). Feeding constructs were obtained from the Ahringer or Openbiosystems libraries and sequenced or newly cloned from *C. elegans* cDNA. HT115 bacteria transformed with feeding vectors were grown at 37°C in LB + ampicillin (100 µg/ml) for 5 hr, induced with 5 mM IPTG for 45 min, plated on NNGM (nematode nutritional growth media) + ampicillin (100 µg/ml) + IPTG (1 mM), and grown overnight at room temperature before adding L4 worms at 24°C for 24–30 hr. For Phos-tag gel experiments, worms were fed starting in the L1 stage and scored for embryonic lethality (*mbk-2(RNAi)*) or P granule phenotype (*pptr-1(RNAi)*).

For sterility counts, at least eight gravid adult worms were allowed to lay embryos for 1 to 2 hr. Adult progeny were scored for empty uteri ('white sterile' phenotype) on a dissecting microscope. To determine the number of germ cells in *meg* larvae, we counted the number of nuclei in GFP::PGL-1-expressing cells, using vulval morphology to stage the larvae.

### Yeast two-hybrid

Yeast two-hybrid assays were performed using DUALhybrid kit (P01004, Dualsystems Biotech, Switzerland). All plasmids were converted to Gateway-compatible vectors (Invitrogen, Carlsbad, CA) and Gateway recombination was used to create N-terminally tagged pLexA-PPTR-1 bait vector, pY3H-PAA-1 bridge vector, and candidate prey vectors. For library screening, yeast transformed with PPTR-1 bait and PAA-1 bridge were used with prey library from Dualsystems consisting of polyA+ cDNA from mixed stage *C. elegans* with $5.7 \times 106$ independent clones (28 µg of prey library). Total transformation efficiency was $4.3 \times 105$ clones per µg DNA, $1.2 \times 107$ total transformants, with 2.1× library coverage.

### GST pull-down

GST fusion proteins were cloned into pGEX6p1 (GE Healthcare, Pittsburgh, PA). MBP fusion proteins were cloned into pJP1.09, a Gateway-compatible pMAL-c2x (*Pellettieri et al., 2003*). Proteins were expressed in *E. coli* BL21 cells overnight at 16°C, following induction with 0.4 mM IPTG. 200 mg of bacterial pellet of GST fusion proteins was resuspended in 10 mM EGTA, 10 mM EDTA, 500 mM NaCl, 0.1% Tween, PBS pH 7.4 with protease and phosphatase inhibitors, lysed by sonication, and bound to GST beads. Beads were washed and incubated with MBP fusion proteins at 4°C for 1 hr in 50 mM Hepes pH 7.4, 1 mM EGTA, 1 mM $MgCl_2$, 500 mM KCl, 10% glycerol, 0.05% NP-40, pH 7.4 with protease and phosphatase inhibitors. After washing, beads were eluted with 10 mM reduced glutathione and eluates were loaded on SDS-PAGE.

### Protein purification and in vitro kinase assay

MBP fusion proteins were cloned into pJP1.09, expressed and partially purified as previously described (*Griffin et al., 2011*). In vitro kinase assays were performed as described (*Cheng et al., 2009*).

### Phos-tag gel

Embryos were harvested from synchronized young adult worms and sonicated in 2% SDS, 65 mM Tris pH 7, 10% glycerol with protease and phosphatase inhibitors. Lysates were spun at 14,000 rpm for 30 min and cleared supernatants treated with 100 U alkaline phosphatase (Roche, Indianapolis, IN). Samples were run in parallel on Phos-tag gels (7% SDS-PAGE with 25 µM Phos-tag and 50 µM $MnCl_2$, Phos-tag from Wako Chemicals, Japan) and 7% SDS-PAGE at 30 mA for 2.5 hr. Gels were washed in

**Table 1.** List of strains used in this study

| Name | Description | Genotype | Reference |
|------|-------------|----------|-----------|
| JH2842 | *pie-1* prom::GFP::PGL-1-*pgl-1* 3'UTR; *pie-1* prom::mCherry::H2B::*pie-1* 3'UTR | *unc-119(ed3) III; axIs1522[pCM4.11]; ltIs37 [pAA64] IV* | **Gallo et al., 2010** |
| JH2843 | *pptr-1* mutant; *pie-1* prom::GFP::PGL-1-*pgl-1* 3'UTR; *pie-1* prom::mCherry::H2B::*pie-1* 3'UTR | *pptr-1(tm3103) V; axIs1522[pCM4.11]; ltIs37 [pAA64] IV* | **Gallo et al., 2010** |
| JH3055 | *meg-3* mutant | *meg-3(tm4259) X* | This study |
| JH3147 | *meg-3* mutant; *pie-1* prom::GFP::PGL-1-*pgl-1* 3'UTR; *pie-1* prom::mCherry::H2B::*pie-1* 3'UTR | *meg-3(tm4259) X; axIs1522[pCM4.11]; ltIs37 [pAA64] IV* | This study |
| JH3225 | *meg-3 meg-4* mutant—see **Table 2** | *meg-3(tm4259) meg-4(ax2026) X* | This study |
| JH3149 | *meg-3 meg-4* mutant; *pie-1* prom::GFP::PGL-1-*pgl-1* 3'UTR; *pie-1* prom::mCherry::H2B::*pie-1* 3'UTR | *meg-3(tm4259) meg-4(ax2026) X; axIs1522[pCM4.11]; ltIs37 [pAA64] IV* | This study |
| JH3148 | *meg-1* mutant; *pie-1* prom::GFP::PGL-1-*pgl-1* 3'UTR; *pie-1* prom::mCherry::H2B::*pie-1* 3'UTR | *meg-1(vr10) X; axIs1522[pCM4.11]; ltIs37 [pAA64] IV* | This study |
| JH3229 | *meg-1 meg-3* mutant | *meg-1(vr10) meg-3(tm4259) X* | This study |
| JH3150 | *meg-1 meg-3* mutant; *pie-1* prom::GFP::PGL-1-*pgl-1* 3'UTR; *pie-1* prom::mCherry::H2B::*pie-1* 3'UTR | *meg-1(vr10) meg-3(tm4259) X; axIs1522[pCM4.11]; ltIs37 [pAA64] IV* | This study |
| JH3156 | *pptr-1; meg-1* mutant; *pie-1* prom::GFP::PGL-1-*pgl-1* 3'UTR; *pie-1* prom::mCherry::H2B::*pie-1* 3'UTR | *pptr-1(tm3103) V; meg-1(vr10) X; axIs1522[pCM4.11]; ltIs37 [pAA64] IV* | This study |
| JH3155 | *pptr-1; meg-3* mutant; *pie-1* prom::GFP::PGL-1-*pgl-1* 3'UTR; *pie-1* prom::mCherry::H2B::*pie-1* 3'UTR | *pptr-1(tm3103) V; meg-3(tm4259) X; axIs1522[pCM4.11]; ltIs37 [pAA64] IV* | This study |
| JH2932 | *mbk-2* null mutant; *pgl-1*::TY1:: eGFP::3xFLAG992C12 | *unc-24(e1172) mbk-2(pk1427) IV / nT1[let-?(m435)](IV;V); ddEX16* | This study |
| YL183 | *meg-1* mutant; *pie-1* prom::GFP:: MEG-1::*pie-1* 3'UTR | *meg-1(vr10) X; GFP::MEG-1* | Leacock and Reinke, 2007 |
| JH3016 | *meg-3* prom::GFP::MEG-3::*meg-3* 3'UTR | *unc-119(ed3); axIS2076[pJW6.01]* | This study |
| JH3064 | *pptr-1* mutant; *meg-3* prom::GFP:: MEG-3::*meg-3* 3'UTR | *pptr-1(tm3103) V; axIS2076[pJW6.01]* | This study |
| JH3019 | *meg-3* prom::GFP::MEG-3::*meg-3* 3'UTR; *pie-1* prom::GFP::PGL-3::*pie-1* 3'UTR | *unc-119(ed3) III; axIS2076[pJW6.01]; axIS2077[pJW2.03]* | This study |
| JH3230 | *pgl-3* mutant; *meg-3* prom::GFP:: MEG-3::*meg-3* 3'UTR | *pgl-3(bn104) V; axIS2076[pJW6.01]* | This study |
| JH3153 | *meg-3* mutant; *meg-3* prom::GFP:: MEG-3::*meg-3* 3'UTR | *meg-3(tm4259) X; axIS2076[pJW6.01]* | This study |
| JH3247 | C-terminal FLAG insertion in genomic *meg-4* locus—see **Table 2** | *meg-4(ax2080)* | This study |
| JH3248 | *meg-4* mutant—see **Table 2** | *meg-4(ax2081)* | This study |

Abbreviations: prom–Promoter, 3' UTR–3' untranslated region, ALL CAPs–coding regions of indicated genes.
All transgenes also contain a wild-type copy of *unc-119* (transformation marker).

transfer buffer with 1 mM EDTA twice for 10 min each and washed in transfer buffer without EDTA twice for 10 min each. Western blot transfer was performed for 1 hr at 4°C onto nitrocellulose membranes. Membranes were blocked and washed in 5% milk, 0.1% Tween-20 in PBS and probed with JL-8 antibody (1:240 dilutions, Clontech).

## Antibody production

Peptide antibodies against MEG-3 were made by Covance (Princeton, NJ) using KLH conjugation in rabbits against the sequence Ac-HAFKKGHKDNKNASC-amide.

**Table 2.** Strains generated by CRISPR/Cas9

| Strain | Allele | Edit | Method | sgRNA (PAM sequence) | Genomic site of edit |
|---|---|---|---|---|---|
| JH3247 | meg-4 (ax2080) | C-terminal 3xFLAG insertion in meg-4 | Homology directed repair with ssODN. Sequence: (homology arms) (cctgtcagatacttgaatgcaaaacgaga atggctgga atctattttgacccaccgagagatcaa)gactacaaaga ccatgacggtgattataaagatcatgatatcgattacaa ggatgacgat gacaag(tgattgtactgatatatatctatt tcatgtcgagtatttt gtattttattcttgttcattgacc) | caatcattgatctctgggt (ggg) | X:1686208 |
| JH3248 | meg-4 (ax2081) | Deletion removing 733 base pairs upstream of the meg-4 start and the first 2565 bases of the gene | NHEJ | gagcgcgaaatagtgtgtg (ggg) tgggaccaaaagcaagaa (tgg) atttatttatggtctgccc (agg) ctgccaggaacttgtaac (ggg) | X:1682223..1685521 |
| JH3325 | meg-3 (tm4259) meg-4 (ax2026) | Deletion of 7 nucleotides starting at nucleotide 37 of meg-4(amino acid 13). The frameshift inserts the amino acids 'EETEKVTALIVEAIWKVHRKIWDTTLV LKNCFIRS' at amino acid 13 followed by a premature stop | NHEJ for meg-4(ax2026) in meg-3(tm4259) | tctctgtttcctctggagtt (tgg) caagttccgttgattccagc(tgg) | X:1682992 |

### In situ hybridization

In situ hybridization of *nos-2* mRNA was performed using fluorescent oligonucleotide probes as described previously (*Voronina et al., 2012*).

### Immunostaining

Gravid adult hermaphrodites were laid on a slide coated with 0.01% poly-L-lysine, and embryos were extruded by squashing with a coverslip. Embryos were frozen on pre-chilled aluminum blocks. Coverslips were removed and slides were incubated in −20°C methanol for 15 min, followed by −20°C acetone for 10 min. Slides were pre-blocked in PBS/0.1% Tween/0.1% BSA (PBT) for 30 min and incubated with primary antibody overnight at 4°C. Primary antibodies were diluted in PBT as follows: K76 (1:10, DSHB, Iowa City, IA), rabbit anti-MEG-3 (1:250, Covance), rabbit anti-MEG-1 (1:100, gift from Valerie Reinke), chicken anti-GLH-2 (1:200, gift from Karen Bennett), anti-FLAG (1:500, Sigma, St. Louis, MO). Secondary antibodies were applied for 2 hr at room temperature.

### Spinning-disc confocal microscopy

Images were acquired using a Zeiss Axio Imager fitted with a Yokogawa spinning disc confocal scanner with Slidebook software (Intelligent Imaging Innovations, Denver, CO) using a 63× objective (embryos) or 40× objective (in utero movies). Embryos were dissected from gravid mothers and mounted on 3% agarose pads in M9 solution at room temperature.

For single time point images, 10 z planes with a z step size of 1 μm, spanning 9 μm, were acquired. Exposure time was 100 ms per plane per color with one exposure acquired in each color before proceeding to the next plane. Granule counts were performed with Slidebook using Mask Segmentation tools and manually verified for each embryo.

For time-lapse movies of embryos, z-stacks (8 z planes, step size 1 μm) were acquired every 8 s. Exposure time was 100 ms per plane per color with one exposure acquired in each color before proceeding to the next plane.

For *in utero* movies, young adult mothers were anesthetized in 0.3 mM levamisole for 15 min prior to mounting on 3% agarose pads. Z-stacks (11 z planes with a z step size of 1 μm) were acquired every 30 s. Exposure time was 100 ms per plane per color with one exposure acquired in each color before proceeding to the next plane.

### Lattice light sheet microscopy

Embryos were dissected from gravid mothers and mounted on 5-mm diameter coverslips in M9 solution at room temperature. Coverslips were pre-cleaned and coated with 2 μl of BD Cell-Tak (BD Biosciences, San Jose, CA) and 2 μl of poly-D-lysine. In the time-lapse dithered mode, 21 z planes with a z step size 0.3 μm, spanning 6 μm, were acquired. Exposure time was 10 ms per plane per color with one exposure acquired in each color before proceeding to the next plane, for a total exposure time of $21 \times 10 \times 2 = 420$ ms for each 3D stack. A 150 ms pause was added between each time point. Dithered-mode stacks were deconvolved in 3D using the Richardson–Lucy algorithm with experimentally measured point spread functions for each color. Granule cross-sectional area was calculated using ImageJ using maximum-projection images from two embryos.

3D-SIM mode images were acquired as in *Chen et al. (2014)*, with 5 phase SIM. For the first image in *Figure 5E*, 71 z planes were acquired with 50 ms exposure per phase and a 0.15 μm z step between planes. For the other images in *Figure 5E*, 71 z planes were acquired with 20 ms exposure per phase and a 0.2 μm z step. SIM images were reconstructed as in *Gao et al. (2012)*. Note that the granules shown in *Figure 5E* span only 7–10 z planes, corresponding to 1–1.75 s acquisition time. We confirmed that during this time, the granules were stationary on the scale of the SIM excitation pattern, allaying any concerns of motion-induced artifacts in the SIM reconstructions.

Photon counts were calculated according to the formula P(photo) = CF × (C − off)/Q(lambda), where CF is the conversion factor (electron/count), C is output count of the selected pixel, off is the background of the camera, and Q(lambda) is a curve based on the wavelength. Signal to noise ratio was calculated as sqrt(P).

## Acknowledgements

We thank Valerie Reinke, Sam Kapelle, Karen Bennett, the National Bioresource Center (Japan), and the CGC (USA) for strains and reagents, the Janelia Farm Visitors Program for hosting Jenn Wang, and

Cliff Brangwynne and the Seydoux lab for discussions. Research in the Seydoux lab is supported by R01HD37047 from the National Institutes of Health. G Seydoux is an Investigator of the Howard Hughes Medical Institute. Research in the Betzig lab is wholly supported by the Howard Hughes Medical Institute.

## Additional information

### Funding

| Funder | Grant reference number | Author |
| --- | --- | --- |
| National Institute of Child Health and Human Development | R01HD37047 | Jennifer T Wang, Jarrett Smith, Alexandre Paix, Bramwell G Lambrus |
| Howard Hughes Medical Institute | | Jarrett Smith, Bi-Chang Chen, Helen Schmidt, Dominique Rasoloson, Deepika Calidas, Eric Betzig, Geraldine Seydoux |

The funders had no role in study design, data collection and interpretation, or the decision to submit the work for publication.

### Author contributions

JTW, All other figures, Conception and design, Acquisition of data, Analysis and interpretation of data, Drafting or revising the article; JS, Figure 3C, Figure 3D, Figure 3 supplement 1 left panel (GFP::MEG-1), Acquisition of data, Analysis and interpretation of data, Drafting or revising the article; B-CC, Videos 5, 6, 7, which were used to make Figure 5D, 5E, Figure 5 - supplements 3B, 3C, 5., Acquisition of data, Analysis and interpretation of data; HS, Figure 4 *meg-4* data, Figure 5 supplement 2A and 2B, Table 2, Acquisition of data, Analysis and interpretation of data, Drafting or revising the article; DR, Figure 3A, Acquisition of data, Analysis and interpretation of data; AP, Figure 4—figure supplement 2A, Acquisition of data, Analysis and interpretation of data; BGL, Generation of strain JH3225, Acquisition of data, Analysis and interpretation of data; DC, Generation of strain JH3247, Acquisition of data, Analysis and interpretation of data; EB, Analysis and interpretation of data, Drafting or revising the article; GS, Corresponding author, Conception and design, Acquisition of data, Analysis and interpretation of data, Drafting or revising the article

### Author ORCIDs

Helen Schmidt, 🆔 http://orcid.org/0000-0002-3449-2790

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
