## [Decision Letter]

Thank you for sending your work entitled “Regulation of RNA granule dynamics by a phosphorylation-sensitive scaffold in *C. elegans*” for consideration at *eLife*. Your article has been favorably evaluated by James Manley (Senior editor), a Reviewing editor, and 4 reviewers, one of whom has agreed to be identified: Roy Parker (Reviewer #1).

The Reviewing editor and the other reviewers discussed their comments before we reached this decision, and the Reviewing editor has assembled the following comments to help you prepare a revised submission. As you will see, all felt that the work was potentially quite interesting but all also raise a number of points that need to be dealt with via revision. In particular all of the reviewers question the conclusion that the MEG proteins serve as a scaffold for P granules. It is essential that you provide convincing evidence in support of this conclusion. Please address this issue as well as the other concerns of the reviewers as thoroughly as possible.

Although we ordinarily collate the reviews into a single narrative, the reviewers felt that you would benefit by a full disclosure of their comments:

Reviewer #1:

This manuscript addresses the mechanisms by which P-granules (maternal mRNA-protein assemblies in *C. elegans*) assemble and disassemble in a temporal and spatially regulated manner. The main contribution of the work is to demonstrate that a phosphorylation-dephosphorylation cycle of the Meg1-4 proteins plays a role in modulating the assembly and disassembly of P-granules. The kinase and phosphatase for this process are identified (building on earlier genetics) and the data generally supports the authors' conclusions, although there are some issues as detailed below.

This review is from Roy Parker and I would be happy to directly clarify the comments for the authors if desired.

A major point of the manuscript (in title and abstract) is that the Meg proteins form a scaffold that modulates P-granule assembly/disassembly. While it is clear that Meg proteins modulate P-granule assembly/disassembly in a phosphorylation dependent manner, the evidence that they form a scaffold is solely that by microscopy they are dynamic and non-uniform within P-granules. Although they could form a scaffold, the current evidence is also open to other interpretations. I suggest they a) change the tone of the manuscript to emphasize that how the Meg proteins affect P-granule structurally is not yet known, and b) remove the scaffold conclusion from the title and the abstract. Alternatively, if the authors want to argue for a scaffolding function per se, they would need additional data to support such a conclusion.

Reviewer #2:

The Wang et. al. paper advances our understanding of regulation of P granule dynamics and partitioning by a kinase, a phosphatase, and cycles of substrate phosphorylation and dephosphorylation. Most of the players have been previously described: MBK-2 kinase is needed for P granule disassembly, PP2A phosphatase (PPTR-1, 2) is needed for P granule assembly/stability, and MEG-1 and MEG-2 are needed for P granule segregation in early embryos and for fertility. By screening for proteins that interact with PPTR-1 in yeast 2-hybrid, Wang et al. additionally identified GEI-11/MEG-3 and MEG-4. They showed that a) MEG-1,3,4 directly interact with PPTR-1,2 and also with the P granule component PGL-1, b) MEG-1 and MEG-3 are phosphorylated by MBK-2 in vitro, c) MEG-1 and MEG-3 are phospho-proteins in vivo and the abundance of their phosphorylated forms is influenced by MBK-2 and PPTR-1. Analysis of double and triple mutants showed that MBK-2 is epistatic to PPTR-1, 2, and the MEGs are epistatic to both. MEG-3, 4 function redundantly to promote P granule assembly. The model is that MBK-2 and MEG phosphorylation promote P granule disassembly in the anterior cytoplasm, and PPTR-1, 2 and MEG dephosphorylation promote P granule assembly in the posterior cytoplasm destined for the germline blastomeres. The MEG proteins are predicted to be largely disordered and based on high resolution imaging of MEG-3 in live embryos using lattice light sheet microscopy, the authors propose the MEGs form a scaffold that surrounds and penetrates the P granules and that regulates P granule dynamics.

This is an impressive “stitch it all together” paper. The genetics are especially compelling. The biochemistry contributes important molecular details, but some of it needs strengthening. The lattice light sheet microscopy images in the figures convey that P granules are heterogeneous and dynamic, and the movies support that MEG-3 forms ribbon-like structures around each granule. Whether that is a “scaffold” is speculative, and should be presented as such.

Comments:

Testing molecular epistasis depends on full knock-out of MBK-2 and PPTR-1, 2 functions. Are the authors confident that double RNAi of mbk-2 and pptr-2 in the mbk-2(RNAi); pptr-1(tm3103) pptr-2(RNAi) triple worked? Might observing an mbk-2 phenotype in the triple be due to poor knock-down of pptr-2? If the authors are confident, I suggest mentioning that mbk-2 is epistatic to pptr-1, 2, as might be expected in a “no phosphorylation” scenario.

Did the authors try to generate any phospho-mimetic aa changes in the MEGs, to see if those cause P granule disassembly, even in the absence of MBK-2?

GFP::PGL-1 is the sole reporter of P granule dynamics in wild type. Does mCherry::PGL-3 display the same dynamics as GFP::PGL-1? Are there other tagged P granule reporters that confirm that the dynamics reported are P granules and not just PGL proteins? Similarly, PGL-1 is the sole reporter of P granule integrity in meg-3 meg-4 mutants. Especially since MEG-3 and MEG-4 interact directly with PGL-1, the authors should verify that P granules are absent in meg-3 meg-4 mutants using additional markers as well.

The lattice light sheet microscopy images in the figures certainly convey that P granules are heterogeneous (the distributions of both MEG-3 and PGL-3) and dynamic, but don't strongly support that MEG-3 forms a scaffold around each granule. Videos 5 and 6 best convey the ribbon-like distribution around the perimeter. Can the authors generate still images that better capture the movies than shown in e.g. Figure 5? How many movies were generated and analyzed? Also, can the authors analyze mCherry::PGL-3 along with GFP::MEG-3 in single time-point images (e.g. Videos 5 and 6) to determine if PGL-3 is located in the spaces between the MEG-3 ribbon zones? Calling the MEGs a scaffold predicts that they would assemble into “granules” in the absence of other P granule components, but in fact MEG granules are disrupted in pgl-1; pgl-3 mutants. I would soften the language from concluding to speculating that the MEGs serve as P granule scaffolds.

There is room for improvement of the analysis of phosphorylation of MEG-3 and MEG-1 shown in Figure 3.

The last two paragraphs of Discussion need to be clarified to accurately reflect the results of this study. In the second to last paragraph, the fact that meg-3 meg-4 adults are 70% fertile, even though these embryos fail to assemble large P granules in germline blastomeres, is used to suggest that P granule assembly is not required for fertility. This is followed by acknowledging that normal looking P granules do actually reform in larval and adult meg-3 meg-4 animals. A more appropriate conclusion would be that P granules ARE required for fertility in adults, but that assembly of large P granules in early embryos is not required to specify the primordial germ cells, similar to their conclusions for pptr-1. In the last paragraph, the authors “suggest that the essential activity of the germ plasm resides not in the germ granules, but in the proteins that scaffold the granules.” It is unclear which of their findings distinguish germ granules from germ granule scaffolds and support their statement.

Reviewer #3:

The manuscript by Wang et. al. “Regulation of RNA granule dynamics by a phosphorylation-sensitive scaffold in *C. elegans*” focuses on the molecular regulation of the assembly and disassembly of *C. elegans* germline P granules. This study reveals that the phosphorylation state of the proteins MEG-1 and MEG-3 is regulated by activity of MBK-2, a kinase homologous to the DYRK kinase shown to regulate stress granule assembly, and the phosphatase PPTR-1/2 PP2A. The authors perform a detailed molecular mechanistic analysis using mutants and/or RNAi, revealing important insights into the epistastic relationships governing P granule assembly. Moreover, the authors utilize a novel super-resolution imaging technique to gain insights into the molecular distribution within P granules. This is an interesting and important study that makes an important contribution to our molecular understanding of P granule assembly, and should therefore definitely be published in *eLife*.

Despite my enthusiasm for much of the work in this paper, I nevertheless have a problem with what I think are over-interpretations of the data, particularly the imaging data.

One of the major concerns I have can be seen clearly in Video 5 (and other movies/images). There is a fair amount of “structure” apparent within the P granules. This is particularly obvious in the largest granule at the center of the image, which looks a bit like popcorn. There is apparently structure within the granules, which exists on a characteristic length scale of ∼200 nm. The problem is that if one looks at regions of the cytoplasm that does not contain granules, there is also structure on this length scale. Why is that? The mottled pattern seen both in granules and in the cytoplasm is quite reminiscent of the patterns one gets after running e.g. a Gaussian bandpass filter through unstructured white noise images, or after running deconvolution algorithms on unstructured images (I just did this for a random, white noise image and the result of applying a Gaussian bandpass filter looks, qualitatively, very similar to the mottled patterns you show; however, I can't seem to figure out how to upload an image with this review...). This is very relevant because for at least some of the light sheet images shown, there is a deconvolution algorithm being employed, where care needs to be taken to avoid artifacts. Do you see structure like that seen in the P granules when you use this technique to image a homogenous solution of dye? If you use this technique to image micron-sized fluorescent spherical particles, for which the embedded dye is presumably uniformly distributed within the particle, does it come out looking like popcorn? (is this also true if the particle is “pre-bleached” so that the fluorophores density is comparable to what one might expect for an average GFP density in the granule?). For these reasons, it is difficult for me to put any weight on arguments made about both the internal structure of the granules, and the idea that they are highly non-spherical on these length scales. Since the authors make such a big deal about the non-homogenity of the granules (e.g. last sentence of the abstract), this seems important to nail down.

As far as I can tell from the movies shown, the mottled patterns are highly dynamic, as one would expect if they were random intensity distributions that were amplified by the imaging technique and post-processing. Together with the observation that the MEG “scaffold” does not form in pgl-1/pgl-3 zygotes, the case for referring to this as a “scaffold”, which implies a static structure on which the rest of the P granule components hang, is a bit of a weak proposition.

Reviewer #4:

This manuscript provides new insight into the process of P granule assembly and disassembly by identifying a set of proteins, the MEGs, whose phosphorylation and dephosphorylation cycle may explain the assembly and disassembly of P granules in the oocyte and early stages of embryonic development. The work is intriguing and most of the data presented are convincing, but the study seems incomplete in several respects. First, it is not clear why the authors chose to analyze some mutants or combinations of mutants and not others. For example, why is the meg-4 mutant omitted? In the analysis of triple mutants, why isn't the phenotype of meg-1/2/3 or meg-2/3/4 shown? The localization analysis is one of the strengths of the paper but is limited to MEG-3 protein; the conclusions would be greatly enhanced by the analysis of other MEGs and, ideally, a combination of two different MEGs differentially labeled with GFP and Cherry. Second, the paper seems to stop short of a critical experiment, analyzing the dynamics of MEG-3 (or other MEG protein) in mbk-2 and pptr-1 mutants or even the distribution in wildtype animals during disassembly. A second major concern is that meg-3 meg-4 mutant animals are fertile despite the fact P1 has very few P granules and none are apparent in P4 (Figure 4), which makes me question the importance of these genes and the P granule assembly process.

Additional issues:

1) Protein-protein interactions are only demonstrated in vitro; the data would be significantly strengthened by demonstration of in vivo interactions through co-IP experiments.

2) The Phos-tag gel data are not particularly convincing. The authors say that they detected “one primary species for both GFP:MEG1 and GFP:MEG3 and several higher mw species that did not resolve well into bands.” These are not at all apparent on the gels shown (or at least there are no higher mw species that appear different between wt and pptr-1 or mbk-2), although a single higher mw isoform of GFP:MEG-3 is visible in the supplemental figure. Also, treatment of MEG-3 by AP in both the mbk-2 mutant, where presumably the protein is not phosphorylated, results in an increase in the intensity of the band comparable to that in the wt. This would suggest that the protein is primarily phosphorylated by one or more kinases other than mbk-2 but the authors seem to ignore this.

3) What are the data that allow the conclusion that the MEGs are scaffolds? They aren't sufficient for P granule assembly and it seems like an over interpretation of the localization data.

---

## [Author Response]

Reviewer #1:

*A major point of the manuscript (in title and abstract) is that the Meg proteins form a scaffold that modulates P-granule assembly/disassembly. While it is clear that Meg proteins modulate P-granule assembly/disassembly in a phosphorylation dependent manner, the evidence that they form a scaffold is solely that by microscopy they are dynamic and non-uniform within P-granules. Although they could form a scaffold, the current evidence is also open to other interpretations. I suggest they a) change the tone of the manuscript to emphasize that how the Meg proteins affect P-granule structurally is not yet known, and b) remove the scaffold conclusion from the title and the abstract. Alternatively, if the authors want to argue for a scaffolding function per se, they would need additional data to support such a conclusion*.

We agree with the reviewer that the use of the word scaffold was too speculative.

Accordingly, we have removed this reference from most of the manuscript except for the discussion, where we present this concept clearly as a hypothesis that will need to be tested further. The hypothesis of a “scaffold” is based on the Following observations:1) the MEGs are required to assemble large PGL granules from smaller unstable PGL granules and 2) MEG‐3 localizes to small assemblies in the cytoplasm and larger assemblies surrounding a PGL core in the P granules and 3) (new data!) during granule disassembly, MEG‐3 and MEG‐4 assemblies Persist longer than the PGL‐1 core. Together these observations suggest the following working model: the MEGs promote granule assembly by forming dynamic scaffolds in the cytoplasm that stabilize and wrap around PGL condensates. See new text in Discussion.

Reviewer #2:

*Testing molecular epistasis depends on full knock-out of MBK-2 and PPTR-1, 2 functions. Are the authors confident that double RNAi of mbk-2 and pptr-2 in the mbk-2(RNAi); pptr-1(tm3103) pptr-2(RNAi) triple worked? Might observing an mbk-2 phenotype in the triple be due to poor knock-down of pptr-2? If the authors are confident, I suggest mentioning that mbk-2 is epistatic to pptr-1, 2, as might be expected in a “no phosphorylation” scenario*.

We are confident that the RNAi treatments worked for three reasons: 1) *mbk‐2(RNAi) pptr‐2(RNAi)* was applied to *pptr‐1(tm3103)* animals and resulted in a Mbk‐2 phenotype consistent with *mbk‐2(RNAi)* working and 2) we applied in parallel control RNAi; *pptr‐2(RNAi)* to *pptr‐1(tm3103)* and obtained the expected Pptr‐1/2 phenotype confirming that the *pptr‐2(RNAi)* treatment also worked. 3) *pptr‐1/2(RNAi)* on the *mbk‐2* null gave the same result. We have clarified this conclusion in the text.

Did the authors try to generate any phospho-mimetic aa changes in the MEGs, to see if those cause P granule disassembly, even in the absence of MBK-2?

We did not make phospho‐mimetic mutants because of the large numbers of potential phosphorylation sites and the difficulty in testing for partial suppression of the *mbk‐2* phenotype (granule number is variable in *mbk‐2* zygotes). We did mutate S574 in MEG‐1 (site shown to be phosphorylated by

MBK‐2 *in vitro* and *in vivo*) to alanine and did not observe a robust phenotype, consistent with the idea that the regulation involves more than one phosphorylation site.

*GFP::PGL-1 is the sole reporter of P granule dynamics in wild type. Does mCherry::PGL-3 display the same dynamics as GFP::PGL-1? Are there other tagged P granule reporters that confirm that the dynamics reported are P granules and not just PGL proteins? Similarly, PGL-1 is the sole reporter of P granule integrity in meg-3 meg-4 mutants. Especially since MEG-3 and MEG-4 interact directly with PGL-1, the authors should verify that P granules are absent in meg-3 meg-4 mutants using additional markers as well*.

We agree with the reviewer that it is important to confirm that the phenotypes we report are not just specific to PGL‐1, but extend to other P granule components. Accordingly, we have now added the following data to the revision:

mCherry::PGL‐3 undergoes the same dynamics as GFP::PGL‐1 in wild‐type embryos (Figure 1—figure supplement 1, column 5). GLH‐2 behaves like PGL‐1 in *meg‐3meg‐4* zygotes (Figure 4—figure supplement 1). We have also confirmed that P granules are absent in *meg‐3meg‐4* zygotes by observing the distribution of *nos‐2* RNA (RNA component of P granules) and PIE‐1 (another RNA binding protein that associates with P granules) (Figure 4—figure supplement 2).

*The lattice light sheet microscopy images in the figures certainly convey that P granules are heterogeneous (the distributions of both MEG-3 and PGL-3) and dynamic, but don't strongly support that MEG-3 forms a scaffold around each granule.*
Videos 5 and 6
*best convey the ribbon-like distribution around the perimeter. Can the authors generate still images that better capture the movies than shown in e.g.*
Figure 5*? How many movies were generated and analyzed? Also, can the authors analyze mCherry::PGL-3 along with GFP::MEG-3 in single time-point images (e.g.*
Videos 5 and 6*) to determine if PGL-3 is located in the spaces between the MEG-3 ribbon zones?*

The strongest evidence that MEG‐3 is present at the perimeter comes from our analysis of 37 granules, where we compared the area occupied by MEG‐3 and

PGL‐3, and found that MEG‐3 occupied a larger area in 34 of the 37 granules, with PGL‐3 entirely contained within the MEG‐3 area in all cases (Figure 5—figure supplement 3).

Unfortunately we were not able to generate SI (highest resolution) images to examine MEG‐3 and PGL‐3 in the same granule, so we do not know whether PGL‐3 is localized in the spaces between the MEG‐3 domains. Our confocal and lower resolution lattice‐sheet (dithered mode) images, however, suggest that this is the case (Figure 5—figure supplement 3).

*Calling the MEGs a scaffold predicts that they would assemble into “granules” in the absence of other P granule components, but in fact MEG granules are disrupted in pgl-1; pgl-3 mutants. I would soften the language from concluding to speculating that the MEGs serve as P granule scaffolds*.

We agree that we have used the word “scaffold” too freely and in the revision we now mention this concept only in the discussion. The observation that the MEGs no longer localize to large granules in the absence of PGL‐1 and PGL‐3 is actually consistent with our model that dynamic MEG scaffolds present throughout the cytoplasm associate with and stabilize PGL condensates. Without the PGL condensates, the dynamic MEG scaffolds remain distributed throughout the cytoplasm.

*There is room for improvement of the analysis of phosphorylation of MEG-3 and MEG-1 shown in*
Figure 3*.*

We have expanded Figure 3—figure supplement 1 to show the complete migration pattern of MEG‐1 and MEG‐3 isolated from wild‐type embryos with and without alkaline phosphatase treatment. We have also included data for MEG‐3 in the *pptr‐1* mutant. At equivalent concentrations of protein and alkaline phosphatase, MEG‐3 shows greater resistance to alkaline phosphatase treatment in the *pptr‐1* background compared to wild‐type.

*The last two paragraphs of Discussion need to be clarified to accurately reflect the results of this study. In the second to last paragraph, the fact that meg-3 meg-4 adults are 70% fertile, even though these embryos fail to assemble large P granules in germline blastomeres, is used to suggest that P granule assembly is not required for fertility. This is followed by acknowledging that normal looking P granules do actually reform in larval and adult meg-3 meg-4 animals. A more appropriate conclusion would be that P granules ARE required for fertility in adults, but that assembly of large P granules in early embryos is not required to specify the primordial germ cells, similar to their conclusions for pptr-1. In the last paragraph, the authors “suggest that the essential activity of the germ plasm resides not in the germ granules, but in the proteins that scaffold the granules.” It is unclear which of their findings distinguish germ granules from germ granule scaffolds and support their statement*.

We completely agree with the reviewer and have clarified our writing to reflect these points. We do not dispute that the perinuclear P granules that form later in development are required for fertility, as has been shown by many studies. Our analyses only address the function of the cytoplasmic P granules that form within the germ plasm of early embryos. We argue that granule assembly in the context of germ plasm of early embryos is NOT essential. This is an important conclusion as many studies assume that RNA granules perform the critical function of the germ plasm. We have reworded the discussion to make this point clearer.

The sentence in the last paragraph is meant to remind readers that, although the MEGs’ role in granule assembly does not appear essential, the MEGs as a group are essential for fertility. We have reworded this statement.

Reviewer #3:

*One of the major concerns I have can be seen clearly in*
Video 5
*(and other movies/images). There is a fair amount of “structure” apparent within the P granules. This is particularly obvious in the largest granule at the center of the image, which looks a bit like popcorn. There is apparently structure within the granules, which exists on a characteristic length scale of ∼200 nm. The problem is that if one looks at regions of the cytoplasm that does not contain granules, there is also structure on this length scale. Why is that? The mottled pattern seen both in granules and in the cytoplasm is quite reminiscent of the patterns one gets after running e.g. a Gaussian bandpass filter through unstructured white noise images, or after running deconvolution algorithms on unstructured images (I just did this for a random, white noise image and the result of applying a Gaussian bandpass filter looks, qualitatively, very similar to the mottled patterns you show; however, I can't seem to figure out how to upload an image with this review...). This is very relevant because for at least some of the light sheet images shown, there is a deconvolution algorithm being employed, where care needs to be taken to avoid artifacts. Do you see structure like that seen in the P granules when you use this technique to image a homogenous solution of dye? If you use this technique to image micron-sized fluorescent spherical particles, for which the embedded dye is presumably uniformly distributed within the particle, does it come out looking like popcorn? (is this also true if the particle is “pre-bleached” so that the fluorophores density is comparable to what one might expect for an average GFP density in the granule?). For these reasons, it is difficult for me to put any weight on arguments made about both the internal structure of the granules, and the idea that they are highly non-spherical on these length scales. Since the authors make such a big deal about the non-homogenity of the granules (e.g. last sentence of the abstract), this seems important to nail down*.

*As far as I can tell from the movies shown, the mottled patterns are highly dynamic, as one would expect if they were random intensity distributions that were amplified by the imaging technique and post-processing*.

We understand the reviewer’s concerns and now include raw data from the SI lattice light sheet analyses in a supplementary figure (Figure 5—figure supplement 5). The new figure shows that:

1) The GFP::MEG‐3 ribbon‐like structure is visible before the reconstruction algorithm is applied (Figure 5—figure supplement 5), showing that reconstruction does not impose structure at the scale of MEG‐3 ribbons.

2) The GFP::MEG‐3 structure is maintained between successive time frames (Figure 5—figure supplement 5), even though movement can also be detected. If the structure were white noise, it would be uncorrelated between frames.

3) The photon counts are at least an order of magnitude greater than noise, ruling out the possibility that the signal is just filtered white noise. (Figure 5—figure supplement 5)

*Together with the observation that the MEG “scaffold” does not form in pgl-1/pgl-3 zygotes, the case for referring to this as a “scaffold”, which implies a static structure on which the rest of the P granule components hang, is a bit of a weak proposition*.

We agree and have removed the mention of a scaffold throughout the text, except

For the discussion where we present this idea as a working hypothesis. We also present new data showing that the MEGs perdure longer in disassembling granules than PGL‐1, consistent with a scaffolding function. We describe our model more clearly in the Discussion.

As mentioned above, we note that the observations that the MEGs no longer localize to large granules in the absence of PGL‐1 and PGL‐3 is actually consistent with our model that dynamic MEG scaffolds present throughout the cytoplasm associate with and stabilize PGL condensates. Without the PGL condensates, the smaller, dynamic MEG scaffolds remain distributed throughout the cytoplasm.

Reviewer #4:

This manuscript provides new insight into the process of P granule assembly and disassembly by identifying a set of proteins, the MEGs, whose phosphorylation and dephosphorylation cycle may explain the assembly and disassembly of P granules in the oocyte and early stages of embryonic development. The work is intriguing and most of the data presented are convincing, but the study seems incomplete in several respects. First, it is not clear why the authors chose to analyze some mutants or combinations of mutants and not others. For example, why is the meg-4 mutant omitted?

We have now added analysis of the *meg‐4* mutant (Figure 4). Our initial RNAi analyses indicated that loss of *meg‐4* alone is not sufficient to cause a P granule phenotype in wildtype, but enhances the phenotype of *meg‐3* mutants. For this reason, we first created a *meg‐4* mutant in the background of the *meg‐3* mutant to create the double (the two genes are closely linked on X). Since then, we have generated a second *meg‐4* mutant in the wild‐type background.

As expected, this mutant has no P granule phenotype. These new data are now included in Figure 4.

In the analysis of triple mutants, why isn't the phenotype of meg-1/2/3 or meg-2/3/4 shown?

Because all the *meg*s are linked on the X chromosome, it is not trivial to create all the different combinations. Analysis of *meg‐1* and *meg‐2* by Leacock and Reinke has shown that, of the two, *meg‐1* plays the primary role, and therefore we used *meg‐1* for the triple mutant combination. We did analyze the quadruple combination and found that loss of *meg‐1* and *meg‐*2 does not affect the meg‐*3 meg‐4* mutant phenotype (Figure 4—figure supplement 3).

*The localization analysis is one of the strengths of the paper but is limited to MEG-3 protein; the conclusions would be greatly enhanced by the analysis of other MEGs and, ideally, a combination of two different MEGs differentially labeled with GFP and Cherry*.

We have done the most extensive analyses on MEG‐3 because it is the only single *meg* that shows a strong P granule phenotype in zygotes. In the revision, we now add new localization data for MEG‐1 and MEG‐4. The data confirm that MEG‐4, like MEG‐1, 2 and 3, localizes to P granules specifically in embryos. The data also demonstrate that MEG‐1, like MEG‐3, only partially overlaps with PGL‐1. See new Figure 5—figure supplement 2 and Figure 5—figure supplement 4.

*Second, the paper seems to stop short of a critical experiment, analyzing the dynamics of MEG-3 (or other MEG protein) in mbk-2 and pptr-1 mutants or even the distribution in wildtype animals during disassembly*.

We have done these analyses (see Figure 5—figure supplement 2). These data indicate that MEG‐3 and MEG‐4 persist longer than PGL‐1 in disassembling granules in wild‐type. We also show that this is the case for GFP::MEG‐3 in *ppt‐1* zygotes. These results indicate that the MEGs exhibit different dynamics than PGL‐1 in P granules. Together with the fact that the MEGs are required for the assembly of large stable PGL granules, these observations are consistent with a possible scaffolding function, but do not prove it. For this reason, while we include the new data in the revision, we have removed mention of “scaffold” from the title and text and only mention this idea as a hypothesis in the Discussion.

*A second major concern is that meg-3 meg-4 mutant animals are fertile despite the fact P1 has very few P granules and none are apparent in P4 (*Figure 4*), which makes me question the importance of these genes and the P granule assembly process*.

We agree with the reviewer and address this point in the last section of the discussion. Our conclusion is that P granule assembly in the germ plasm of early embryo is NOT essential for fertility. MEG activity, however, is required for fertility since *meg‐1meg‐2* and *meg‐1meg‐3meg‐4* mutants are 100% sterile.

We suggest therefore that MEGs performs a function required for fertility that is NOT dependent on P granule assembly. This is an important conclusion as granule assembly is often assumed to be an essential function of the germ plasm.

*Additional issues*:

*1) Protein-protein interactions are only demonstrated in vitro; the data would be significantly strengthened by demonstration of in vivo interactions through co-IP experiments*.

The proposed interaction between the MEGs and PPTR‐1 is based on four independent lines of evidence: interaction in the yeast two hybrid system, GST pull downs using partially purified proteins, *in vivo* phosphorylation patterns visualized on Phostag gels, and genetic epistasis.

We also showed that the MEGs interact with PGL‐1 in GST pull‐down assay. As indicated by the reviewer, the MEG‐PGL interaction will need to be confirmed *in vivo,* since we do not provide any additional evidence for that interaction*.* However, that interaction is not the main focus of this study and none of our conclusions depend on that finding. We can remove those data if the editor prefers.

*2) The Phos-tag gel data are not particularly convincing. The authors say that they detected “one primary species for both GFP:MEG1 and GFP:MEG3 and several higher mw species that did not resolve well into bands.” These are not at all apparent on the gels shown (or at least there are no higher mw species that appear different between wt and pptr-1 or mbk-2), although a single higher mw isoform of GFP:MEG-3 is visible in the supplemental figure*.

We agree with the reviewer that it is difficult on the Phos‐tag gels to see all the phosphorylated forms of MEG‐1 and MEG‐3. This is consistent with the fact that these proteins have large numbers of serines and threonines and likely exist in many different isoforms, each with a unique migration behavior on the Phos‐tag gels.

We include larger gels in Figure 3—figure supplement 1 that clearly show the presence of at least one phosphoisoform. Since we cannot visualize all isoforms on gels, we based all of our quantitative analyses on the non‐phosphorylated form of MEG‐1 and MEG‐3, which runs as a clear single band in all the experiments. By comparing the abundance of this band on Phos‐tag versus non‐Phos‐tag gels, we estimate the degree of phosphorylation in the different extracts and use these estimates to COMPARE degrees of phosphorylation between wild‐type and *mbk‐2* and *pptr‐1* mutants.

These experiments were repeated at least 4 times, and always showed the same results: the degree of phosphorylation is highest in *pptr‐1* lysates and lowest in *mbk‐2* lysates.

*Also, treatment of MEG-3 by AP in both the mbk-2 mutant, where presumably the protein is not phosphorylated, results in an increase in the intensity of the band comparable to that in the wt*.

That experiment was not done with a *mbk‐2* mutant (which is lethal and cannot be grown in large enough quantities for biochemistry), but with *mbk‐2(RNAi)* embryos which may retain some *mbk‐2* activity and therefore some phosphorylation.

Also, as we note in the text, it is very possible that other kinases phosphorylate the MEGs. Our only claim is that MBK‐2 is one of the kinases that influence the degree of MEG phosphorylation.

*This would suggest that the protein is primarily phosphorylated by one or more kinases other than mbk-2 but the authors seem to ignore this*.

We agree and have included the following in the text: “we do not know whether MBK‐2 and PP2A PPTR‐1/2 are the only regulators of MEG‐1 and MEG‐3 phosphorylation, or whether other kinases and phosphatases are also involved.”

*3) What are the data that allow the conclusion that the MEGs are scaffolds? They aren't sufficient for P granule assembly and it seems like an over interpretation of the localization data*.

We agree and now only mention the scaffold idea as a working hypothesis in the Discussion.